# Phased Training for LLM-powered Text Retrieval Models Beyond Data Scaling

**Xin Zhang**[1,2]**, Yanzhao Zhang, Wen Xie**[1]**, Dingkun Long, Mingxin Li**
**Penjun Xie, Meishan Zhang**[1]*****Wenjie Li**[2]**, Min Zhang**[1]
[1]Harbin Institute of Technology, Shenzhen  [2]The Hong Kong Polytechnic University
zhangxin2023@stu.hit.edu.cn {zhangmeishan,zhangmin2021}@hit.edu.cn
Project: github.com/vec-ai/lychee-embed    Model: hf.co/vec-ai/lychee-embed

## Abstract

Current efforts in building large language models (LLMs) based general-purpose text retrieval models primarily focus on architectural design and training data scaling. However, significant challenges remain in effectively modeling diverse retrieval tasks and domains, including multi-task conflict, data imbalance, and training efficiency. To address these challenges, we propose a novel phased training framework for text retrieval, featuring: (1) robust foundation modeling with core relevance data, (2) progressive specialization through modular task adaptation, and (3) knowledge fusion via weight interpolation based model merging. This framework simultaneously optimizes both embedding and reranking models through a unified architecture. We also present an efficient yet scalable data synthesis pipeline to expand training data, based on open-source LLMs. These synthetic data can be efficiently incorporated into the phased training framework, enhancing model performance. We identify five distinct types of retrieval tasks, *i.e.,* basic relevance retrieval, code retrieval, tool retrieval, complex instruction-based retrieval, as well as reasoning-intensive retrieval, conducting extensive experiments. Our method achieves the best performance across MTEB and various retrieval benchmarks of the five task types. Further analysis demonstrates the effectiveness and efficiency of our proposed training framework and data synthesis pipeline.

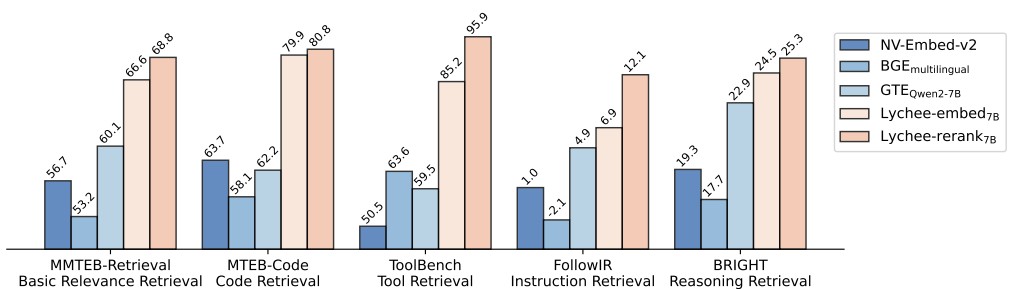

Figure 1: The performance of our Lychee models and baselines on different retrieval tasks.

## 1 Introduction

Text retrieval serves as a foundational technology for numerous downstream applications, including web search (Huang et al., 2020; Liu et al., 2021) and question answering (Karpukhin et al., 2020). Modern competitive text retrieval systems typically adopt a multi-stage pipeline: dense vector retrieval via embedding models for initial candidate recall, followed by cross-encoder rerankers for fine-grained relevance scoring (Zhao et al., 2024).

---

*Corresponding author.

Recently, the emergence of Large Language Models (LLMs) has profoundly reshaped retrieval tasks. LLMs introduce dual advancements: (1) enhanced semantic understanding capabilities that enable superior embedding and reranking models, and (2) novel application paradigms such as Retrieval-Augmented Generation (RAG) and Agent Systems, which demand new specialized retrieval capabilities beyond traditional semantic matching, including tool or API retrieval (Xu et al., 2024), instruction-based retrieval (Weller et al., 2025a), and reasoning-intensive retrieval (Su et al., 2025). These emerging scenarios require models to not only recognize basic relevance but also comprehend functional compatibility, procedural logic, and task-specific constraints.

To build such general-purpose retrieval models, existing approaches primarily use LLMs as backbone models, scaling training data through public sources and LLM-driven synthetic dataset (Wang et al., 2024; Lee et al., 2025a;b). The training strategy still inherits BERT-era methodologies, where diverse data are naïvely mixed or sampled to perform contrastive learning. However, previous schemes suffer from critical challenges in building versatile models for various basic and specialized retrieval tasks. In the aspect of data synthesis, prior research relies on expensive commercial LLM services for data generation and filtering to ensure the quality (Wang et al., 2024; Lee et al., 2025b), which severely restrict the efficiency and scalability of data scaling. Moreover, the directly mixing of data in a single-step training approach presents several obvious issues: (1) conflict in multi-task retrieval learning (Yu et al., 2020; Li et al., 2024), (2) data imbalance across tasks, domains and languages (Wang et al., 2024), and (3) inefficient training and developing process (§4.3).

To address these challenges, we propose a new phased training framework with adaptive model merging, designed to progressively enhance retrieval capabilities. Our framework first categorizes training datasets into clusters based on task similarity and data volume, then builds the model in a phased manner: **(1)** establishing foundational relevance competencies across tasks on sampled warm-up data, **(2)** separately training specialized task models for each cluster, avoiding the impact of data imbalance and task conflicts, **(3)** merging all task models into a composite one with adaptive parameter interpolation (Li et al., 2024) to fuse multi-task knowledge, and **(4)** fine-tuning the merged model to optimize adaptability across tasks. This framework offers a more efficient way compared to the complex and expensive process of tuning data sampling ratios in traditional multi-task training. In addition, we introduce a cost-effective data synthesis pipeline leveraging role-aware LLM prompting with open-source LLMs, generating high-quality multi-domain query-document pairs without reliance on commercial APIs. More importantly, these synthetic data can be quickly integrated into the final model through the separate training and merging paradigm in the proposed framework, enhancing the performance across multiple retrieval tasks.

In this work, we identify five types of retrieval tasks (§4.1), including basic relevance retrieval, code retrieval, tool retrieval, complex instruction-based retrieval, and reasoning-intensive retrieval. Built upon powerful Qwen2.5 (Yang et al., 2024) LLMs, our framework produces a suite of general text embedding and reranking models, **Lychee series**, achieving new state-of-the-art results across multiple evaluation benchmarks of the five types (§4.2). We further demonstrate the effectiveness and efficiency of our training framework and data synthesis pipeline through extensive analysis (§4.3). Our contributions are threefold:

• **Phased Training Paradigm**: We propose a multi-stage training strategy with adaptive model merging that enables efficient knowledge transfer across tasks, outperforming traditional single-stage approaches in extensibility and final performance.

• **Scalable LLM-Driven Data Synthesis**: We present a scalable, cost-effective method for generating high-quality retrieval data, which is validated to match or exceed effectiveness of existing data synthesis pipelines.

• **Open-Source Release**: We will open-source our general-purpose Lychee embedding and reranking models and training protocols to facilitate advancements in LLM-powered retrieval systems.

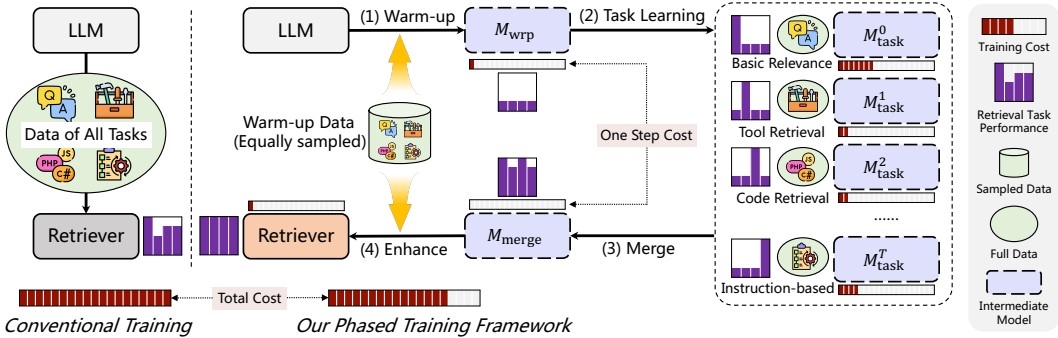

Figure 2: Illustration of conventional training and our phased training framework. The conventional approach typically mixes all different types of data together and performs single-stage multi-task training, which faces challenges of data imbalance and task conflicts. Moreover, any adjustment requires re-running the entire training process. To address these challenges, our framework first learns each retrieval task independently, followed by model merge and enhance phases, enabling a more efficient training process (analysis in §4.3).

## 2 Related Work

**Text-centric Retrieval**    Text-centric information retrieval tasks have evolved substantially in recent years. Traditional text retrieval tasks primarily focus on information-seeking scenarios, matching user queries with relevant passages containing direct answers (Nguyen et al., 2016; Thakur et al., 2021). This can be extended to multilingual retrieval across language boundaries (Zhang et al., 2023b) and code retrieval for finding relevant code snippets (Husain et al., 2019). Beyond information-seeking, with the development of LLMs, more specialized retrieval scenarios have emerged. For example, (1) instruction following (Su et al., 2023; Weller et al., 2025a) where the search needs to follow specific instructions, (2) tool retrieval that focuses on finding appropriate tools or APIs for specific functionalities (Zhang et al., 2023a; Xu et al., 2024), and (3) reasoning-intensive retrieval that requires complex reasoning to better understand the query (Xiao et al., 2024a; Su et al., 2025). While these tasks have typically been addressed separately, our work explores a unified approach capable of handling diverse scenarios while maintaining competitive performance.

**Multi-stage Retrieval with Embedding and Reranking**    Retrieval tasks are primarily addressed through a multi-stage approach, typically consisting of *first-stage retrieval* for efficiently selecting candidates (e.g., top 1,000) from a large corpus, followed by *reranking* for precise relevance assessment (Zhao et al., 2024). Modern first-stage retrieval has shifted to dense retrieval approaches (Karpukhin et al., 2020), where an embedding model encodes queries and documents into dense vectors for semantic matching. The reranking stage commonly employs transformer-based cross-encoder models for fine-grained scoring of query-candidate pairs (Zhao et al., 2024), enabling richer interaction between query and candidate compared to first-stage retrievers. With the widespread adoption of RAG applications, pre-trained *general-purpose* embedding and reranking models have garnered significant attention (Wang et al., 2022; Chen et al., 2024; Zhang et al., 2024). In particular, LLM based approaches have consistently pushed the boundaries of retrieval performance (Wang et al., 2024; Muennighoff et al., 2025; BehnamGhader et al., 2024; Lee et al., 2025a; Li et al., 2025). The success of these models can be attributed to their effective single-stage large-scale training on data from diverse domains and tasks. However, simple data combination can lead to multi-task conflict issues (Yu et al., 2020), which limits the model training (Sturua et al., 2024). In this work, we aim to improve LLM-based general-purpose embedding and reranking models, proposing a novel phased training framework with model merging (Li et al., 2024) to tackle the challenge in multi-task/domain training.

**LLM-driven Retrieval Data Synthesis**    Beyond architectural and training improvements, scaling training data is crucial for building powerful general-purpose retrieval models.

However, the scarcity of authentic data presents a significant challenge (Villalobos et al., 2024). To address this, using LLMs to synthesize retrieval data has proven to be an effective solution, such as synthesizing multi-task and multilingual retrieval data (Wang et al., 2024) and long-context retrieval pairs (Chen et al., 2024). Furthermore, some researchers propose using synthetic retrieval data as evaluation sets, offering a more cost-effective (Rahmani et al., 2025) and dynamic model assessment (Chen et al., 2025). However, existing retrieval data generation pipelines often rely heavily on expensive proprietary LLMs (Wang et al., 2024) or involve complex workflows (Chen et al., 2025), making the data synthesis process either prohibitively expensive or inefficient. In this paper, we present a novel role-based data generation pipeline that enables efficient and cost-effective data scaling.

## 3 Method

### 3.1 Implementation of Embedding and Reranking

Our embedding and reranking models are both designed within an instruction retrieval framework, where each query $q$ is accompanied by a task instruction $I$ to guide the retrieval of the relevant document $d^+$ (*a.k.a.,* positive). Each training instance comprises $I$, $q$, $d^+$ and a list of irrelevant documents $[d_1, d_2, ..., d_n]$ (*i.e.,* negatives).

**Embedding** We utilize the LLM backbone to encode the input text, where the final hidden state corresponding to the [EOS] token is extracted as the final text embedding. Concretely, the query embedding is computed as $e_q = LLM(\{I, q, [EOS]\})[-1]$ and the doc embedding is $e_d = LLM(\{d, [EOS]\})[-1]$. The model is trained by the following contrastive loss:

$$\mathcal{L}_{cl} = -\log \frac{\exp(\cos(e_q, e_d^+)/\tau)}{\sum_{j=1}^{N} \exp(\cos(e_q, e_{d_j}^-)/\tau) + \exp(\cos(e_q, e_d^+)/\tau)} \, , \tag{1}$$

where $e_q$, $e_d^+$, $e_{d_j}^-$ represents the embedding of query, positive document and the $j$-th negative document separately, and $N$ is the number of hard negatives. $\tau$ is the temperature parameter.

**Reranking** We transform the relevance judgment task into a binary classification problem. Specifically, we use a softmax function on the probabilities of the model outputs for "yes" and "no" to obtain the final ranking score. We optimize the following loss:

$$\begin{aligned} \mathcal{P}(q, d) &= \langle \text{prefix} \rangle \text{ Instruct: } I \text{ Query:} q \text{ Document: } d \\ L_{rank} &= -\log p(l | \mathcal{P}(q, d)) \end{aligned} \tag{2}$$

where $p(\cdot | *)$ denotes the LLM probability output by the LLM, and the $\langle \text{prefix} \rangle$ is a fixed prompt for relevance judgment: "Given the instruction and query, judge whether the document is relevant to the query (yes/no).". The label $l$ is "yes" for positive documents and "no" for negatives.

### 3.2 Phased Training Framework

To develop a general model for multiple retrieval tasks, a straightforward approach is to combine the training data from all tasks into a single dataset. However, this naïve method often leads to data imbalance issues, as the volume of available training data typically varies significantly across different tasks. For instance, general relevance retrieval tasks usually have substantially more training data compared to specialized tasks such as tool retrieval. While ensuring balanced task contribution during training is crucial for achieving strong performance across all tasks, finding the optimal data composition through experimental search can be computationally expensive. Moreover, simply combining data from different tasks may lead to multi-task conflict issues (Yu et al., 2020; Li et al., 2024).

To address these challenges, we propose a phased training framework (Figure 2). Given datasets from multiple tasks, we first classify them into $T$ clusters $\{\mathcal{C}_1, \ldots, \mathcal{C}_T\}$ based on the amount of available data and task similarity. The framework consists of four steps:

**(1) Warm-up:** We sample small and equal proportions of data from each cluster to build a warm-up training data $\mathcal{D}_{\mathrm{wrp}}$. Then we train an initial model $M_{\mathrm{wrp}}$ on these data, establishing fundamental relevance capabilities across all tasks.

**(2) Task Learning:** For each cluster $\mathcal{C}_i$, we continue train $M_{\mathrm{wrp}}$ on the full-scale data to obtain the specialized task model $M_{\mathrm{task}}^i$. This step ensures that each model develops expertise in its respective task cluster.

**(3) Merge:** All task models $\{M_{\mathrm{task}}^0, \dots, M_{\mathrm{task}}^T\}$ are merged into a single composite model $M_{\mathrm{merge}}$. The merging method is detailed in §3.3.

**(4) Enhance:** Finally, to optimize the model's adaptability across different tasks while maintaining task-specific competencies, we fine-tune $M_{\mathrm{merge}}$ on the warm-up data $\mathcal{D}_{\mathrm{wrp}}$.

### 3.3 Adaptive Model Merging

Our method is inspired by findings from Li et al. (2024) and employ the SLERP merging approach[1], which utilizes spherical linear interpolation for model merging and a sampled tiny dataset for hyperparameter search. Given two models (*i.e.*, $M_i$ and $M_j$) trained from an identical base model $M_0$, SLERP first calculates task vectors ($v_i$ and $v_j$) which represent the parameter differences between the respective trained models and the original base model, *e.g.*, $v_i = \theta(M_i) - \theta(M_0)$. Let $\alpha_{ij}$ denote the angle between the task vectors of $v_i$ and $v_j$. The merged task vector $v_{ij} = f_{\mathrm{slerp}}(v_i, v_j)$ is computed as:

$$f_{\mathrm{slerp}}(v_i, v_j) = \frac{\sin((1-t)\alpha_{ij})}{\sin(\alpha_{ij})} v_i + \frac{\sin(t\alpha_{ij})}{\sin(\alpha_{ij})} v_j \,, \tag{3}$$

where $t$ is a hyperparameter. Extending this approach to $N$ models, the final task vector $V$ is derived as follows:

$$V = \sum_{i=1}^{N} f_{slerp}(v_k, V_{i-1}) \,, \tag{4}$$

where $V_{i-1}$ is the merged task vector of $N-1$ models. Thus, the final model parameters is $\theta = \theta(M_0) + \lambda \times V$, where $\lambda$ is a hyperparameter controlling task vector scaling. The search for optimal merging hyperparameters could be posed as an optimization problem:

$$(\{\hat{t}_i\}_{i=1}^N, \hat{\lambda}) = \mathrm{argmin}_{(\{t_i\}_{i=1}^N, \lambda)} \left( \frac{1}{|D_{\mathrm{t}}|} \sum_{I \in D_{\mathrm{t}}} \mathcal{L}(I; \theta) + \mu\lambda \right) \,, \tag{5}$$

where $\mu$ is a hyperparameter used to prevent overfitting in this optimization problem, and $D_{\mathrm{t}}$ is a tiny sample set for efficient hyperparameter search.

### 3.4 Scalable LLM-Driven Data Synthesis

We present a simple but effective data synthesize method by LLM (*i.e.*, Qwen2.5-72B-Instruct Yang et al. (2024)), which involves a two-step generation process guided by simulated user roles and query categories (Figure 3). This approach aims to diversify and enrich the training data for downstream tasks. Our data synthesis begins with comprehensive raw corpus collection from diverse sources, including news articles, Wikipedia articles, academic papers, and technology publications (Appendix §A.3). The variety of source material helps ensure the generated data covers a broad spectrum of topics and writing styles.

**Role Candidates Selection** Inspired by AIR-Bench (Chen et al., 2025) and Persona Hub (Ge et al., 2024), we assign specific roles to each document to simulate the persona of a user that might pose queries about the document. This injection of user perspective enhances the diversity and realism of the synthetic queries. Specifically, we leverage a retrieval model[2] to identify the top 5 role candidates for each document from Persona Hub.

---

[1]Refer to https://github.com/Digitous/LLM-SLERP-Merge
[2]https://hf.co/Alibaba-NLP/gte-Qwen2-1.5B-instruct

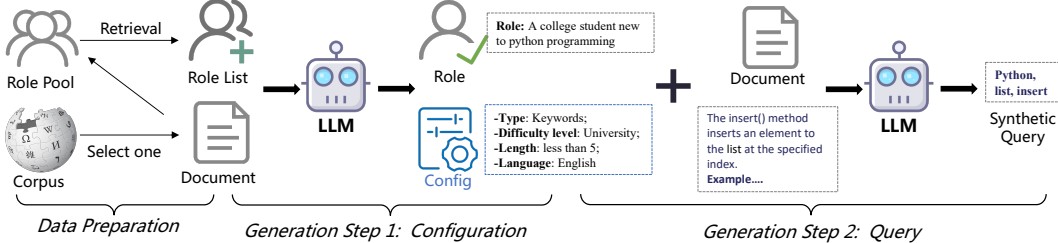

Figure 3: Our two-step generation pipeline for synthesizing retrieval data.

**Generation Step 1 - Configuration** In this step, the sampled document and its role candidates are provided to a prompt, instructing the LLM to output the most appropriate role-configuration combination for query generation. The query configuration consists of 4 dimensions: (1) `Type`: we define five query types, {*Keywords, Factual, Summarization, Yes/No, Background*}; (2) `Difficulty Level`: we set three levels of difficulty, {*High School, University, PhD*}; (3) `Length`: we specify different length option lists[3] for different question types; and (4) `Language`: English or Chinese. The combination of these dimensions allows for fine-grained control over the characteristics of the generated queries.

**Generation Step 2 - Query Generation** The selected document and generated query configuration serve as the input for this step. The LLM is prompted with the following structured template to generate queries that adhere to the defined type, difficulty, length, and language constraints, maintaining consistency and relevance to the original document.

```
You need to start from the perspective of the following {Role}, generate a
query based on the {Document} and the {Requirements}.
**Role:** ...
**Document:** ...
**Requirements:**
- Type:{};    - Difficulty level:{};
- Length:the length of the generated sentences should be {} words;
- Language:the language of generation should be {};
```

**Filtering** We implement a filtering process with two checks to ensure the quality and relevance of the generated queries: (1) **Format validation**: Any queries that do not conform to the predefined JSON format are discarded during the parsing phase; (2) **Consistency filtering**: We employ an embedding model[2] to assess the relevance between generated queries and their corresponding documents. Queries that fail to retrieve their associated document within the top-20 results are filtered out. This quality assurance step ensures that only high-quality, relevant queries are retained in the final dataset.

## 4 Experiments

### 4.1 Setup

**Evaluation Tasks** We explore five types of retrieval tasks: (1) Basic Relevance Retrieval, we set three categories: English, Chinese, and Multilingual, and evaluate on MTEB (Muennighoff et al., 2023), CMTEB (Xiao et al., 2024b), MMTEB (Enevoldsen et al., 2025) and MLDR (Chen et al., 2024), respectively; (2) Code Retrieval, we evaluate on MTEB-Code (Enevoldsen et al., 2025), which composed entirely of code retrieved data; (3) Tool Retrieval, on ToolBench[4] (Qin et al., 2024); (4) Complex Instruction Retrieval, on FollowIR (Weller et al., 2025a); and (5) Reasoning Intensive Retrieval, on BRIGHT (Su et al., 2025). For MTEB,

---

[3]For example, `Keywords` corresponds the length list {*less than 5, less than 7, 7 to 13*}, and `Summarization` corresponds the length list {*13 to 21, 20 to 50, more than 80 but less than 100*}.

[4]We use the processed version by Zhang et al. (2023a) for retrieval.

| Task Type (5) | Description | Data Cluster (4) | #Train | Evaluation Benchmark |
|---|---|---|---|---|
| BR | Basic Relevance Retrieval | EN & ZH | 4M | MTEB* (Muennighoff et al., 2023) CMTEB* (Xiao et al., 2024b) |
| | | Multilingual | 0.5M | MMTEB* (Enevoldsen et al., 2025) MLDR (Chen et al., 2024) |
| CR | Code Retrieval | Code & Tool | 1.2M | MTEB-Code (Enevoldsen et al., 2025) |
| TR | Tool Retrieval | | | ToolBench (Qin et al., 2024) |
| CIR | Complex Instruction-based Retrieval | CIR | 1M | FollowIR (Weller et al., 2025a) |
| RIR | Reasoning Intensive Retrieval | | | BRIGHT (Su et al., 2025) |

Table 1: Our text-centric retrieval experiment setting. For reasoning intensive retrieval, we do not have training data as there are no appropriate datasets publicly available. *We evaluate on the full set for embedding models and the retrieval subset for reranking.

| | | BR | | | | CR | TR | CIR | RIR |
|---|---|---|---|---|---|---|---|---|---|
| Model | Param | MTEB | CMTEB | MMTEB | MLDR | MTEB-Code | ToolBench | FollowIR | BRIGHT |
| BGE$_{multilingual}$ | 9.24B | 69.88$^\alpha$ | 68.44$^\alpha$ | 61.25$^\gamma$ | 49.10$^\gamma$ | 62.04$^\gamma$ | 63.65$^\gamma$ | -2.13$^\gamma$ | 17.68$^\gamma$ |
| NV-Embed-v2 | 7.85B | 72.31$^\alpha$ | - | 56.25 | - | 63.74$^\gamma$ | 50.54$^\gamma$ | 1.04 | 19.28$^\gamma$ |
| GritLM-7B | 7.24B | 66.8$^\alpha$ | - | 60.93 | - | 73.6$^\sigma$ | 35.42 | 3.45 | 20.63 |
| E5$_{mistral-7b}$ | 7.11B | 66.6$^\alpha$ | 59.92 | 60.28 | - | 69.2$^\sigma$ | 31.79 | -0.62 | 17.54 |
| GTE$_{Qwen2-7B}$ | 7.62B | 69.88 | 71.62 | 62.51 | 56.53$^\gamma$ | 62.17$^\gamma$ | 59.48$^\gamma$ | 4.94 | 22.89 |
| **Lychee-embed$_{Qwen2.5-7B}$** | 7.62B | **73.00** | **73.43** | **65.12** | **57.90** | **79.93** | **85.16** | **6.91** | **24.50** |
| GTE$_{Qwen2-1.5B}$ | 1.54B | 67.19 | 67.12 | **59.47** | 52.11$^\gamma$ | 61.98$^\gamma$ | 62.57$^\gamma$ | 0.74 | 18.47$^\gamma$ |
| **Lychee-embed$_{Qwen2.5-1.5B}$** | 1.54B | **68.39** | **69.77** | 58.43 | **53.85** | **72.54** | **86.35** | **5.74** | **19.47** |
| BGE-M3 (Dense) | 0.56B | 59.84$^\gamma$ | 61.79$^\gamma$ | 59.54$^\gamma$ | **52.50**$^\alpha$ | 58.22$^\gamma$ | 58.45$^\gamma$ | -3.11 | 11.94$^\gamma$ |
| Jina-v3 | 0.57B | 65.52$^\alpha$ | 63.07$^\gamma$ | **58.37**$^\alpha$ | 40.71$^\gamma$ | 58.85$^\gamma$ | 59.64$^\gamma$ | -1.34 | 11.34$^\gamma$ |
| **Lychee-embed$_{Qwen2.5-0.5B}$** | 0.49B | **65.73** | **69.26** | 56.81 | 51.44 | **71.06** | **85.43** | **2.24** | **13.25** |

Table 2: Evaluation for embedding models. Scores are reported from the MTEB leaderboard by default to ensure consistency and comparability across evaluations. $^\alpha$Taken from the corresponding paper or model card. $^\gamma$Our run. $^\sigma$From Enevoldsen et al. (2025).

CMTEB, MMTEB, we follow their scoring methods and report the overall average. For MLDR, ToolBench, BRIGHT, we use nDCG@10. For FollowIR, we use its defined p-MRR.

**Training Data** We select public training datasets for each retrieval task type, then manually categorize into *four* clusters based on language and task relevance for our training framework. Table 1 presents the distribution of different data types used throughout the model training process (detailed datasets in Appendix Table 5). Due to limited public data for tool retrieval and overlap with code retrieval, they are combined for training. Moreover, we employ 2M LLM synthesized data (the pipeline is elaborated in §3.4, statistics in Table 8) to enhance the retrieval training.

**Implementation** To support these diverse tasks with multilingual input, we choose the Qwen2.5 LLMs (Yang et al., 2024) as our backbone. We adopt Qwen2.5 models in three sizes: 0.5B, 1.5B, and 7B, which yield embeddings of 1024, 1536, and 3584 dimensions, respectively. We employ LoRA (Hu et al., 2022) with the rank of 64 and the alpha of 32. In training, we use the AdamW (Loshchilov & Hutter, 2019) optimizer with the dynamic batch size strategy to improve the training efficiency of different length texts (details refer to Appendix Table 6). We train the models for approximately 10,000 steps during the Warm-up and Enhance stage. The number of steps for the Task learning stage varies depending on the clustering data size: around 200,000 steps for the English and Chinese (EN & ZH) cluster, 50,000 steps for the Multilingual, 100,000 steps for Code&Tool, and 20,000 steps for CIR. In the Merge stage, we search for the merging hyper-parameters using roughly 100 steps. The learning rate is set to 1e-4 with a warm-up ratio of 0.1 in all training.

| Model | Param | BR | | | | CR | TR | CIR | RIR |
| | | BEIR | CMTEB-R | MMTEB-R | MLDR | MTEB-Code | ToolBench | FollowIR | BRIGHT |
|---|---|---|---|---|---|---|---|---|---|
| **Lychee-embed**Qwen2.5-1.5B | 1.54B | 57.16 | 72.98 | 59.28 | 53.85 | 72.54 | 86.35 | 5.74 | 19.47 |
| Jinamultilingual-reranker-v2-base | 278M | 54.61 | 70.18 | 54.43 | 50.32 | 46.32 | 67.80 | -0.69 | 16.69 |
| BGEreranker-v2-m3 | 568M | 55.36 | 71.82 | 57.13 | 60.80 | 50.81 | 62.52 | -0.06 | 15.87 |
| BGEreranker-v2-gemma | 9.24B | 60.81 | 71.74 | **69.80** | 49.10 | 68.63 | 68.14 | -2.13 | 17.68 |
| **Lychee-rerank**Qwen2.5-0.5B | 0.49B | 56.03 | 69.79 | 58.09 | 59.74 | 69.86 | 86.79 | 6.29 | 11.58 |
| **Lychee-rerank**Qwen2.5-1.5B | 1.54B | 59.56 | 76.37 | 62.47 | 64.09 | 78.03 | 90.82 | 7.38 | 16.92 |
| **Lychee-rerank**Qwen2.5-7B | 7.62B | **62.85** | **79.06** | 68.77 | **64.50** | **80.78** | **95.50** | **12.11** | **25.31** |

Table 3: Evaluation results for reranking models. We use the retrieval subsets of MTEB, CMTEB and MMTEB, which are BEIR, CMTEB-R and MMTEM-R. The rest are all retrieval tasks. All scores are our runs based on the retrieval top-100 results from the first row.

## 4.2 Main Results

**Embedding** In Table 2, we present the evaluation results for embedding models. We compare our models with three types of representative baselines: **(1)** LLM-powered English-centered versatile embedders, *i.e.,* NV-Embed-v2 (Lee et al., 2025a), GritLM-7B (Muennighoff et al., 2025); **(2)** LLM-based multilingual embedders, *i.e.,* BGEmultilingual (Chen et al., 2024), GTEQwen2 (Li et al., 2023), and E5mistral-7b (Wang et al., 2024); and **(3)** strong BERT-based multilingual models (for comparation with our 0.5B model), *i.e.,* BGE-M3 (Chen et al., 2024) and Jina-v3 (Sturua et al., 2024). Our models achieve the best performance across all tasks, including all levels of retrieval (basic relevance and specialized), demonstrating the effectiveness of our approach. Detailed results of each benchmark are provided in the Appendix, along with ablation studies on our training framework and data in §4.3.

**Reranking** In Table 3, we report the evaluation results for reranking models, where we use the retrieval subset of MTEB (*i.e.,* BEIR (Thakur et al., 2021)), CMTEB (CMTEB-R) and MMTEB (MMTEB-R) for evaluation. To make a fair comparison, we take the retrieval top-100 results from Ours-EmbedderQwen2.5-1.5B as candidates to all rerankers. Similar to the embedding evaluation, we compare our rerankers with: **(1)**: LLM-powered general-purpose rerankers, *i.e.,* BGEreranker-v2-gemma; (2) BERT-based rerankers, *e.g.,* BGEreranker-v2-m3. Similar to the embedding evaluation, out models demonstrate superior performance compared to previous models of comparable size.

## 4.3 Analysis

**Training Efficiency** There are two commonly adopted baseline training strategies in current practice: **(1)** naïve training, which directly combines training data from different tasks; **(2)** naïve training with up-sampling, which increases the probability of appearance for tasks with smaller data scales before conducting native multi-task training. Since they both are single-stage training, any adjustments to data or hyper-parameters would require to run the entire training process again, which is computationally expensive. Our framework, in contrast, learns different data clusters separately, allowing for individual adjustments with greater efficiency. Based on the experimental settings in Table 1, we analyze and compare the efficiency gaps. Following the conventional setting of training for one epoch, we use the amount of training data consumed as a *proxy* for training cost. Assuming we make one modification to the CIR data (1M) during training. For **naïve training**, the cost of one training iteration is $(4 + 0.5 + 1.2 + 1) = 6.7$, resulting in a total cost of 13.4 for two training iterations. For **up-sampling** with a coefficient of 1.3 for small-scale data, one training iteration costs $4 + (0.5 + 1.2 + 1) * 1.3 = 7.51$, leading to a total training cost of 15.02. In **our framework**, Warm-up and Enhance steps each consume 0.1, while Merge is virtually cost-free. The task learning phase consumes 6.7, making the initial training cost 6.9. Since we need to readjust CIR, the additional costs of 1 and 0.1 are required for re-running the CIR task learning and Enhance, bringing the total cost to $6.9 + 1 + 0.1 = 8$, which is significantly lower than both 13.4 and 15.02. This demonstrates the efficiency advantages of our framework. For reference, we also list the time required for the 1.5b embedding model when training on

| Row | Setting | BR | | | | CR | TR | CIR | RIR |
|---|---|---|---|---|---|---|---|---|---|
| | | BEIR | CMTEB-R | MMTEB-R | MLDR | MTEB-Code | ToolBench | FollowIR | BRIGHT |
| 1 | Lychee-embed$_{\text{Qwen2.5-0.5B}}$ | 56.03 | 70.69 | 57.70 | 49.10 | 71.06 | 85.43 | 2.24 | 13.25 |
| 2 | w/o Enhance (*i.e.*, $M_{\text{merge}}$) | 53.99 | 70.10 | 56.71 | 48.26 | 72.01 | 84.97 | 2.25 | 13.16 |
| 3 | w/o Merge (split Task models) | 56.12 | 70.12 | 56.43 | 49.77 | 71.10 | 85.43 | 1.56 | 17.39 |
| 4 | w/o Task Learning ($M_{\text{wrp}}$) | 50.69 | 67.38 | 52.38 | 45.76 | 51.20 | 64.98 | 0.69 | 10.39 |
| 5 | w/o synthetic data | 53.30 | 69.20 | 55.78 | 44.79 | 70.26 | 85.45 | 1.99 | 13.12 |
| 6 | Naïve | 52.91 | 70.10 | 46.71 | 40.26 | 71.05 | 67.12 | 1.15 | 12.16 |
| 7 | Naïve w/ up-sampling | 55.47 | 70.50 | 57.28 | 48.65 | 70.79 | 85.63 | 2.16 | 13.39 |

Table 4: Ablation study. For brevity, we report results from our 0.5B embedding model, with other model variants showing consistent trends. Row 6 and 7 are trained on the same data with row 1, including the synthetic data. The naïve refers to simply combining all data to train the model. The up-sampling method involves increasing the probability of appearance for data subsets with smaller scales.

32 GPUs, where our method takes about 7 days, the naïve method takes about 11.5 days, and the up-sampling takes about 13.5 days.

To conclude, our framework yields similar strong performance compared to basic up-sampling strategy, but allows more efficient model iteration and adjustment.

**Ablation study of the training framework. Table 4** We conduct ablation experiments on the embedding model to evaluate the performance of intermediate models, analyzing the impact of each step in our training framework. Our framework consists of four steps, producing three types of intermediate models: the warm-up model $M_{\text{merge}}$, task-specific models, and the merged model $M_{\text{wrp}}$. **First**, during the *warm-up* step, we train the LLM on small-scale data to acquire a fundamental retrieval model $M_{\text{wrp}}$. As shown in row 4 of Table 4, $M_{\text{wrp}}$ performs relatively weak across various retrieval tasks. **Then**, in the *task learning* step, we derive specialized models for each task from $M_{\text{wrp}}$. Row 3 demonstrates that all task-specific models achieved significant improvements compared to $M_{\text{wrp}}$ (row 4). **Next**, we *merge* multiple task models into a single model $M_{\text{merge}}$, whose performance (row 2) generally remains comparable to task-specific models (row 3), with slight degradation in some cases. **Finally**, after the *enhance* step, we obtain the final model. Comparing it (row 1) with $M_{\text{merge}}$ (row 2), we observe further improvements across all tasks. **Additionally**, we train the model using two baseline strategies without our framework (rows 6 & 7). Their inferior performance compared to our final model validates that our training framework effectively enhances the development of multi-task retrieval models. For comparisons with other model merging methods, we recommend referring to Li et al. (2024), which provides a comprehensive analysis of various merging methods on text embedding.

**Data synthesis quality evaluation. Table 4 & Figure 4** We proposed an efficient data generation method (§3.4) to create extensive synthetic data for both Chinese and English Basic Relevance Retrieval. We validate its effectiveness from two perspectives. **First**, we perform a direct ablation by removing synthetic data from the training process. As shown in row 5 of Table 4, the model trained without synthetic data shows lower performance compared to our final model (row 1), demonstrating the beneficial impact of synthetic data. **Second**, we conduct controlled experiments comparing different data generation pipelines, including AIR-Bench (Chen et al., 2025) and Promptagator (Dai et al., 2023). Using the same 400k corpus from MS MARCO, we generate synthetic datasets of equal size using three different methods, train three small models, and evaluate performance on BEIR. As illustrated in Figure 4, our method achieves the best performance, outperforming the two baselines, while providing higher efficiency and more diverse synthetic data.

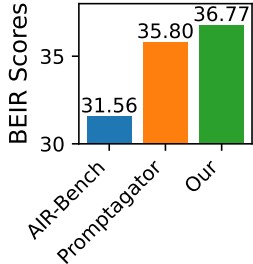

Figure 4: Comparison of different synthesis methods. We generate data from the same corpus, train small models, and evaluate on BEIR.

# 5 Conclusion

In this work, we aim to develop enhanced LLM-based general-purpose text embedding and reranking models, focusing on five retrieval tasks: basic relevance retrieval, code retrieval, tool retrieval, complex instruction retrieval, and reasoning-intensive retrieval. We propose a novel phased training framework to address multi-task learning conflicts and data imbalance through model merging and task-specific learning. Additionally, we introduce an efficient data synthesis method for scaling, employing a two-step generation process based on user roles and query configurations. With these advancements, we build a series of models of varying sizes based on robust multilingual LLMs. Our models demonstrate superior performance across eight benchmarks covering all five tasks, underscoring the effectiveness of our approach. We conduct ablation studies to confirm that our training framework significantly enhances multi-task retrieval learning and our data synthesis method is more effective than previous approaches.

## Acknowledgments

We sincerely thank the anonymous reviewers for their valuable feedback and suggestions. This work receives partial support from the Natural Science Foundation of China (under Grant 624B2048) and Research Grant Council of Hong Kong (PolyU/15209724), and the Shenzhen Science and Technology Program (under Grant ZDSYS20230626091203008).

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

# Appendix

## A   Implementation Details

### A.1   Training Data

To build strong general-purpose retriever models, we collect various public datasets for training and employ our efficient data synthesis pipeline to generate around 2M synthetic data for basic relevance retrieval. Table 5 presents the detailed datasets.

| Cluster | Instances | |
|---------|-----------|---|
| En & Zh | 4M | MS MARCO(2016), Natural Questions (NQ)(2019), TriviaQA(2017), HotpotQA(2018), SQuAD(2016), FEVER(2018), MEDI(2023), AllNLI (2021), DuReader (2022), mMARCO-zh (2021), T2-Ranking(2023), CmedQAv2(2018), SimCLUE[5], Multi-CPR(2022), Our synthesized data (2M) |
| Multilingual | 0.5M | Mr. TyDi (2021), MIRACL (2023b), MLDR (2024) |
| Code & Tool | 1.2M | CodeSearchNet(2019) , MetaTool(2024), APIGen(2024) ,ToolBench(2024) |
| CIR | 1M | Promptriever (2025b) |

Table 5: Details of the training data.

### A.2   Training Setup

In the Warm-up stage, we sample 25,000 instances from each cluster, totaling 100, 000 instances for training. During the Task Learning stage, the model is trained using the complete dataset from each cluster. In the Model-Merging stage, we sample 1,000 instances from all clusters. For the Enhance stage, we fine-tune the model using the same 100,000 instances from the Warm-up stage.

We use a dynamic batch size strategy in the training process to speed up the training with long-context instances. Table 6 shows the batch size setting for different models.

For the embedding model, we add MRL loss (Kusupati et al., 2022) in the Enhance step to enable the model to generate elastic embeddings, providing the flexibility to adapt the embedding dimensionality based on computational constraints or task requirements.

| length | BS | S-BS | BS(R) | S-BS(R) |
|--------|-----|------|-------|---------|
| 0-500 | 512/64/32 | 128/64/32 | 128/96/64 | 128/64/32 |
| 500-1000 | 64/32/24 | 128/64/8 | 96/48/32 | 128/64/8 |
| 1000-2000 | 8/4/4 | 8/4/4 | 12/6/4 | 8/4/4 |
| 2000-3000 | 4/4/4 | 6/3/2 | 8/6/4 | 6/3/2 |
| 3000-8000 | 4/2/2 | 4/2/1 | 6/4/2 | 4/2/1 |

Table 6: Batch size (BS) and sub batch size (S-BS) of different length for embedding (E) and reranker (R) model across the 0.5B, 1.5B, and 7B parameter models

### A.3   Data Generation

In this subsection, we describe the details of our efficient data synthesis pipeline and present a concrete two-step generation example in Figure 5.

**Corpus collection** We employ a doc2query-style method to synthesize relevance retrieval data. To ensure the synthesized data spans various domains and subjects, we collect passages in both Chinese and English from diverse fields. For the Chinese data, we refer

| Query Type | Definition | Length |
|---|---|---|
| Keywords | The query is composed of a separate set of words that are the center of the passage. Such as: Federal learning, AI security; | {less than 5, less than 7, 7 to 13} |
| Factual | Ask questions about a sentence in the passage, such as: what,where,when,why; | {5 to 20, 13 to 27, more than 20 but less than 50} |
| Summarization | Summarize the main content of the passage, such as: This passage tells about...; | {13 to 21, 20 to 50, more than 80 but less than 100} |
| Yes/No | The query can be answered by yes or no, such as: Is... ?; | {5 to 20, 13 to 27, more than 20 but less than 50} |
| Background | This type of query consists of a background, where the character describes a background (his state, what is being done) and the problem he has encountered, and the provided passage can help him. For example, an intern who is owed wages describes the situation and asks if he can get paid, and the relevant legal documents can help him; | {5 to 20, 25 to 80, more than 80} |

Table 7: Five query types, definitions and the optional length

to IndustryCorpus[6], sampling passages from 17 domains such as Programming, Law, and Sports. For English, we select passages from datasets including finwebedu, wiki_6M, arxiv, ccnews, and cnn. The passages are categorized by length and evenly sampled across five intervals: 0–512, 512–1024, 1024–2048, 2048–4096, and 4096–8192 tokens. We collect 2M passages in both Chinese and English, ensuring a comprehensive corpus for study.

**Role preparation** We choose the open-source PersonaHub[7] dataset as our candidate set for personas, which contains 375,000 detailed role types. These include roles such as *'A scientist who studies the ocean and its ecosystems, focusing on the aphotic zone and its unique creatures. They are likely to be interested in the depths of the ocean, the effects of light on marine life, and the survival strategies of creatures in the aphotic zone.'* Using a dense retrieval model, we identify the Top-5 personas most likely to be interested in a given passage. Subsequently, we prompt the LLM to select the most suitable persona from these five candidates. The LLM adopts the perspective of this chosen persona to pose a question about the passage. This approach yields more detailed and varied outcomes than directly generating associated personas, thereby enhancing the diversity of the synthesized data.

**Query type definition** To further enhance the diversity of synthesized data, we define five query types: Keywords, Factual, Summarization, Yes/No, Background. Each query type is assigned a different optional length, with specific definitions and lengths detailed in Table 7.

**Generation details** We use the Qwen-2.5-72B-Instruct[8] model for the generation in both steps, deploying it locally using the vLLM acceleration framework. For generation parameters, we set the temperature to 0.7, top_p to 0.8, and top_k to 50. Additionally, to further accelerate inference, we enable prefix caching (enable_prefix_caching=True) to cache task descriptions and examples within the prompt. Synthesizing queries for two million passages takes approximately 3,000 GPU hours on A100 80G units.

## A.4 Data statistics

We provide statistics on the synthesized query data in Table 8. The table includes the number of queries generated for each query type, as well as their average lengths.

---

[6]https://hf.co/datasets/BAAI/IndustryCorpus
[7]https://hf.co/datasets/proj-persona/PersonaHub
[8]https://hf.co/Qwen/Qwen2.5-72B-Instruct

Given a **Passage** and **Character**, select the appropriate option from three fields: Character, Question type, Difficulty, and return the result in JSON format.
First, select the Character who are likely to be interested in the Passage from the candidates. Then select the Question type that the Character might ask about the Passage; Finally, choose the Difficulty of the possible question based on the Passage, the Character, and the Question type.

Question type: *{Keywords | Factual | Summarization | Yes or No | Background}*
Difficulty: *{High School | University | PhD}*

Here are a few examples for your reference:
... *(omitted some for space)*

Now it's the **Passage** and the **Character** you have to deal with, your output must always be a JSON object only.
**Passage**
204. (a) All wages, …earned by any person in any employment are due and payable twice during each calendar month,; ... (Amended by Stats. 2019, Ch. 508. (SB 698))*(omitted some for space)*
**Character List**
[" A new employee, fresh out of school, with a strong interest in workplace drama. ", …," a migrant worker who works hard to improve his family life ", "a secondary vocational student who just started working in a California hotel "," a young employee who is struggling in the workplace "]

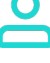

{
    "Character": " a secondary vocational student who just started working in a California hotel ",
    "Question type": "Background",
    "Difficulty": "PhD"
}

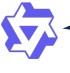

-------------------- new session --------------------

You need to start from the perspective of the following **Character**, generate a question or summary based on the **Passage** and the **Requirment**

Here are a few examples for your reference:
... *(omitted some for space)*

Now you need to generate according to the **Character**, **Passage** and **Requirement**.
**Character**
a secondary vocational student who just started working in a California hotel
**Passage**
204. (a) All wages, …earned by any person in any employment are due and payable twice during each calendar month,; ... (Amended by Stats. 2019, Ch. 508. (SB 698))*(omitted some for space)*
** Requirment**
  -Type: *Background*;
  -Difficulty level: *PhD*;
  -Length: the length of the generated sentences should be *20 to 50* words;
  -Languange: the language in which the results are generated should be *English*;

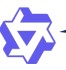

"I am a secondary vocational student, and now I am working in a hotel in California. During the probation period, I have been working for 7 days, but my salary is monthly. If I resign now, can I still get the 7 days' salary?"

Figure 5: A example of synthesizing query data.

| Query Type | Keywords | Factual | Summarization | Yes/No | Background |
|---|---|---|---|---|---|
| **Num.** | 397,079 | 400,339 | 400,111 | 402553 | 399918 |
| **Avg. Token Length** | 9.02 | 17.29 | 88.14 | 14.15 | 90.31 |

Table 8: Statistics of the synthesized queries.

| MTEB English #Datasets ($\rightarrow$) | Param. | Avg. 56 | Avg.$^{\mathrm{T}}$ 7 | Class. 12 | Clust. 11 | PairC. 3 | Rerank. 4 | Retr. 15 | STS 10 | Summ. 1 |
|---|---|---|---|---|---|---|---|---|---|---|
| BGE$_{\mathrm{multilingual}}$$^{\alpha}$ | 9.24B | 69.88 | 66.09 | 88.08 | 54.65 | 85.84 | 59.72 | 59.24 | 83.88 | 31.20 |
| NV-Embed-v2$^{\alpha}$ | 7.85B | 72.31 | 67.97 | 90.37 | 58.46 | 88.67 | 60.65 | 62.65 | 84.31 | 30.7 |
| GritLM-7B$^{\alpha}$ | 7.24B | 66.80 | 64.14 | 79.5 | 50.6 | 87.2 | 60.5 | 57.4 | 83.4 | 30.4 |
| E5$_{\mathrm{mistral-7b}}$$^{\alpha}$ | 7.11B | 66.60 | 64.31 | 78.5 | 50.3 | 88.3 | 60.2 | 56.9 | 84.6 | 31.4 |
| GTE$_{\mathrm{Qwen2-7B}}$ | 7.62B | 69.88 | 66.30 | 86.58 | 56.92 | 85.90 | 61.42 | 58.86 | 83.06 | 31.35 |
| **Lychee-embed$_{\mathrm{Qwen2.5-7B}}$** | 7.62B | 73.00 | 68.69 | 90.48 | 59.30 | 88.96 | 63.55 | 62.97 | 85.38 | 30.23 |
| GTE$_{\mathrm{Qwen2-1.5B}}$ | 1.54B | 67.19 | 64.43 | 82.53 | 48.75 | 87.52 | 59.98 | 58.29 | 82.81 | 31.17 |
| **Lychee-embed$_{\mathrm{Qwen2.5-1.5B}}$** | 1.54B | 68.39 | 65.19 | 86.07 | 52.80 | 86.80 | 61.03 | 57.15 | 82.45 | 30.09 |
| BGE-M3$^{\gamma}$ (Dense) | 0.56B | 59.84 | 56.44 | 73.09 | 37.27 | 84.50 | 55.29 | 48.73 | 81.46 | 31.55 |
| Jina-v3 | 0.57B | 65.52 | 62.76 | 82.58 | 45.27 | 84.01 | 58.13 | 53.87 | 85.80 | 29.71 |
| **Lychee-embed$_{\mathrm{Qwen2.5-0.5B}}$** | 0.49B | 65.73 | 63.27 | 82.65 | 49.80 | 84.43 | 59.02 | 54.49 | 81.23 | 31.29 |

Table 9: Results on MTEB English (Muennighoff et al., 2023). We compare models from the online leaderboard. $^{\alpha}$Taken from the corresponding paper or model card. $^{\gamma}$Our run. Avg.$^{\mathrm{T}}$ means average by task type.

| C-MTEB #Datasets ($\rightarrow$) | Param. | Avg. 35 | Avg.$^{\mathrm{T}}$ 6 | Class. 9 | Clust. 4 | PairC. 2 | Rerank. 4 | Retr. 8 | STS 8 |
|---|---|---|---|---|---|---|---|---|---|
| BGE$_{\mathrm{multilingual}}$$^{\alpha}$ | 9.24B | 68.44 | 69.82 | 74.11 | 59.30 | 86.67 | 68.28 | 73.73 | 56.87 |
| E5$_{\mathrm{mistral-7b}}$ | 7.11B | 59.92 | 60.51 | 72.96 | 52.30 | 66.31 | 61.38 | 61.75 | 48.34 |
| GTE$_{\mathrm{Qwen2-7B}}$ | 7.62B | 71.62 | 72.19 | 75.77 | 66.06 | 81.16 | 69.24 | 75.70 | 65.20 |
| **Lychee-embed$_{\mathrm{Qwen2.5-7B}}$** | 7.62B | 73.43 | 74.91 | 76.95 | 80.80 | 82.43 | 68.35 | 76.37 | 63.58 |
| GTE$_{\mathrm{Qwen2-1.5B}}$ | 1.54B | 67.12 | 67.79 | 72.53 | 54.61 | 79.50 | 68.21 | 71.86 | 60.05 |
| **Lychee-embed$_{\mathrm{Qwen2.5-1.5B}}$** | 1.54B | 69.77 | 69.94 | 74.54 | 79.13 | 68.03 | 66.24 | 73.00 | 58.71 |
| BGE-M3$^{\gamma}$ (Dense) | 0.56B | 61.79 | 62.94 | 67.99 | 45.84 | 73.98 | 74.64 | 65.27 | 49.94 |
| **Lychee-embed$_{\mathrm{Qwen2.5-0.5B}}$** | 0.49B | 69.26 | 70.52 | 72.13 | 76.14 | 78.08 | 64.96 | 71.07 | 60.70 |

Table 10: Results on C-MTEB (Xiao et al., 2024b) (MTEB Chinese). We compare models from the online leaderboard. $^{\alpha}$Taken from the corresponding paper or model card. $^{\gamma}$Our runs. Avg.$^{\mathrm{T}}$ means average by task type.

# B  Evaluation Results

For MTEB, CMTEB, MMTEB, MTEB-code, and FollowIR, we employ the evaluation protocol from the mteb toolkit. For MLDR, ToolBench, and BRIGHT, we use our implementations. We list the retrieval instructions for all evaluation subsets in Table 19.

## B.1  Embedding Benchmarks

Following the practice of build versatile embedding models, we evaluation our embedders and baseline models on the Massive Text Embedding Benchmark (Muennighoff et al., 2023) (Table 9) as well as its Chinese (Xiao et al., 2024b) (Table 10) and Multilingual (Enevoldsen et al., 2025) (Table 11) extensions. There retrieval subset are used for reranker evaluation.

## B.2  Retrieval Benchmarks

In this section, we present the detailed results on all retrieval benchmarks evaluated in this paper, *i.e.*, MTEB-Retrieval/BEIR (Table 12), CMTEB-Retrieval (Table 13), MMTEB-Retrieval (Table 14), MLDR (Chen et al., 2024) (Table 15), MTEB-Code (Table 16), FollowIR (Weller et al., 2025a) (Table 17) and BRIGHT (Su et al., 2025) (Table 18).

| M-MTEB #Datasets (→) | Param. | Avg. 132 | Avg.$^T$ 9 | Btxt. 13 | Class. 43 | Clust. 17 | InstR. 3 | MultiC. 5 | PairC. 11 | Rerank. 6 | Retr. 18 | STS 16 |
|---|---|---|---|---|---|---|---|---|---|---|---|---|
| BGE$_{multilingual}$$^α$ | 9.24B | 61.25 | 51.25 | 67.85 | 67.67 | 49.60 | -7.12 | 21.18 | 80.83 | 57.80 | 53.19 | 73.47 |
| NV-Embed-v2 | 7.85B | 56.25 | 49.64 | 57.84 | 57.29 | 41.38 | 1.04 | 18.63 | 78.94 | 63.82 | 56.72 | 71.10 |
| GritLM-7B | 7.24B | 60.93 | 53.83 | 70.53 | 61.83 | 50.48 | 3.45 | 22.77 | 79.94 | 63.78 | 58.31 | 73.33 |
| E5$_{mistral-7b}$ | 7.11B | 60.28 | 53.18 | 70.58 | 60.31 | 51.39 | -0.62 | 22.20 | 81.12 | 63.82 | 55.75 | 74.02 |
| GTE$_{Qwen2-7B}$ | 7.62B | 62.51 | 56.00 | 73.92 | 61.55 | 53.36 | 4.94 | 25.48 | 85.13 | 65.55 | 60.08 | 73.98 |
| **Lychee-embed$_{Qwen2.5-7B}$** | 7.62B | 65.12 | 58.20 | 74.69 | 63.58 | 56.26 | 6.92 | 26.73 | 84.23 | 66.75 | 66.20 | 75.21 |
| GTE$_{Qwen2-1.5B}$ | 1.54B | 59.47 | 52.75 | 62.51 | 58.32 | 52.59 | 0.74 | 24.02 | 81.58 | 62.58 | 60.78 | 71.61 |
| **Lychee-embed$_{Qwen2.5-1.5B}$** | 1.54B | 58.43 | 52.43 | 59.93 | 57.67 | 51.85 | 5.74 | 22.45 | 78.85 | 64.38 | 59.28 | 69.79 |
| BGE-M3$^γ$ (Dense) | 0.56B | 59.54 | 59.28 | 79.11 | 60.35 | 41.79 | -3.11 | 20.10 | 80.76 | 62.79 | 54.59 | 74.12 |
| Jina-v3 | 0.57B | 58.37 | 50.75 | 65.25 | 58.77 | 46.4 | -1.34 | 18.38 | 79.27 | 57.09 | 55.76 | 77.13 |
| **Lychee-embed$_{Qwen2.5-0.5B}$** | 0.49B | 56.81 | 50.78 | 48.12 | 54.17 | 51.07 | 3.74 | 21.97 | 79.12 | 62.32 | 57.71 | 68.94 |

Table 11: Results on MMTEB (Enevoldsen et al., 2025) (Multilingual MTEB). We compare models from the online leaderboard. $^α$Taken from the corresponding paper or model card. $^γ$Denote our runs. Avg.$^T$ means average by task type.

| BEIR | Avg. | Argu-Ana | Cli-mate-Fever | CQA-Dup-Stack | DB-Pedia | Fever | FiQA | Hotpot-QA | MS MAR-CO | NF-Corpus | NQ | Quora | Sci-docs | Sci-fact | Touche-2020 | Trec-Covid |
|---|---|---|---|---|---|---|---|---|---|---|---|---|---|---|---|---|
| BGE$_{multilingual}$$^α$ | 59.24 | 77.37 | 39.37 | 47.94 | 51.37 | 90.38 | 60.04 | 83.26 | 45.71 | 38.11 | 71.45 | 90.04 | 26.93 | 72.05 | 30.26 | 64.27 |
| NV-Embed-v2 | 63.21 | 70.07 | 45.39 | 58.40 | 53.50 | 93.75 | 65.73 | 85.48 | 45.63 | 44.96 | 73.57 | 89.03 | 21.90 | 80.11 | 31.78 | 88.85 |
| E5$_{mistral-7b}$ | 57.07 | 61.65 | 38.35 | 45.99 | 48.89 | 87.84 | 56.81 | 75.72 | 43.06 | 38.58 | 63.54 | 89.61 | 16.31 | 76.42 | 26.27 | 87.03 |
| GTE$_{Qwen2-7B}$ | 58.86 | 56.57 | 45.88 | 42.66 | 52.42 | 95.11 | 62.03 | 73.08 | 45.98 | 40.60 | 67.00 | 90.10 | 23.48 | 79.06 | 30.57 | 80.37 |
| **Lychee-embed$_{Qwen2.5-7B}$** | 62.97 | 80.02 | 49.98 | 52.00 | 55.13 | 94.55 | 71.11 | 82.20 | 43.75 | 43.67 | 73.86 | 90.71 | 38.84 | 82.32 | 23.51 | 62.85 |
| **Lychee-rerank$_{Qwen2.5-7B}$** | 62.85 | 80.38 | 49.83 | 48.77 | 55.90 | 92.27 | 74.83 | 85.17 | 47.15 | 36.48 | 70.43 | 89.31 | 38.29 | 82.03 | 25.68 | 73.45 |
| GTE$_{Qwen2-1.5B}$ | 58.29 | 69.72 | 42.91 | 44.76 | 48.69 | 91.57 | 54.70 | 68.95 | 43.36 | 39.34 | 63.99 | 89.65 | 24.98 | 78.44 | 27.89 | 85.38 |
| **Lychee-embed$_{Qwen2.5-1.5B}$** | 57.15 | 69.25 | 45.04 | 47.33 | 50.19 | 90.99 | 53.71 | 75.50 | 41.57 | 38.72 | 64.91 | 89.84 | 26.27 | 78.33 | 26.69 | 58.95 |
| **Lychee-rerank$_{Qwen2.5-1.5B}$** | 59.56 | 69.15 | 42.63 | 47.79 | 52.41 | 93.90 | 54.20 | 85.02 | 45.67 | 38.85 | 67.28 | 88.52 | 29.97 | 77.92 | 25.26 | 74.88 |
| BGE-M3 (Dense) | 48.34 | 53.95 | 29.52 | 39.09 | 39.80 | 81.38 | 41.30 | 69.44 | 38.32 | 31.43 | 60.60 | 88.57 | 16.39 | 64.36 | 22.63 | 55.59 |
| Jina-v3 | 53.17 | 43.28 | 42.36 | 42.59 | 41.00 | 89.05 | 47.35 | 64.67 | 40.82 | 36.63 | 64.23 | 89.09 | 19.92 | 72.53 | 26.30 | 77.74 |
| **Lychee-embed$_{Qwen2.5-0.5B}$** | 54.49 | 79.00 | 41.46 | 40.83 | 43.88 | 91.46 | 47.69 | 69.96 | 40.24 | 34.44 | 59.64 | 89.50 | 22.88 | 74.20 | 20.53 | 61.61 |
| **Lychee-rerank$_{Qwen2.5-0.5B}$** | 56.03 | 60.30 | 40.82 | 43.04 | 48.95 | 91.83 | 46.04 | 82.67 | 44.08 | 31.07 | 60.47 | 88.62 | 21.52 | 72.42 | 29.16 | 79.29 |

Table 12: BEIR benchmark (Thakur et al., 2021). We report nDCG@10 scores.

| CMTEB-R | Avg. | Cmedqa | Covid | Du | Ecom | MMarco | Medical | T2 | Video |
|---|---|---|---|---|---|---|---|---|---|
| BGE$_{multilingual}$ | 73.73 | 42.21 | 77.46 | 90.46 | 69.30 | 84.70 | 62.02 | 86.26 | 77.40 |
| E5$_{mistral-7b}$ | 61.75 | 34.23 | 73.11 | 87.04 | 45.95 | 74.84 | 52.83 | 80.68 | 45.34 |
| GTE$_{Qwen2-7B}$ | 75.71 | 48.69 | 81.04 | 87.44 | 71.15 | 85.16 | 65.59 | 87.73 | 78.84 |
| **Ours$_{Qwen2.5-7B}$** | 76.37 | 48.64 | 81.07 | 88.83 | 73.99 | 85.24 | 66.83 | 88.03 | 78.32 |
| **Lychee-rerank$_{Qwen2.5-7B}$** | 79.06 | 48.46 | 92.33 | 91.00 | 73.77 | 88.31 | 68.69 | 88.88 | 79.58 |
| GTE$_{Qwen2-1.5B}$ | 71.86 | 46.97 | 80.79 | 89.40 | 62.51 | 83.01 | 58.65 | 85.47 | 68.11 |
| **Lychee-embed$_{Qwen2.5-1.5B}$** | 73.00 | 43.16 | 79.02 | 88.74 | 69.11 | 82.03 | 62.76 | 83.93 | 75.26 |
| **Lychee-rerank$_{Qwen2.5-1.5B}$** | 76.37 | 45.32 | 90.84 | 89.09 | 69.42 | 85.59 | 64.78 | 84.63 | 78.24 |
| BGE-M3 (Dense) | 65.27 | 33.76 | 77.57 | 83.97 | 58.44 | 77.28 | 54.22 | 81.38 | 56.93 |
| Jina-v3 | 68.54 | 35.84 | 78.55 | 83.13 | 60.89 | 79.69 | 56.64 | 83.16 | 70.43 |
| **Lychee-embed$_{Qwen2.5-0.5B}$** | 71.07 | 42.71 | 77.43 | 87.10 | 64.55 | 80.00 | 59.10 | 82.73 | 74.96 |
| **Lychee-rerank$_{Qwen2.5-0.5B}$** | 69.79 | 35.11 | 90.92 | 80.15 | 63.20 | 79.06 | 55.83 | 78.90 | 75.21 |

Table 13: The retrieval performance on C-MTEB (Xiao et al., 2024b). We report nDCG@10 scores.

| MMTEB-R | Avg. | AILA-Statutes | Argu-Ana | Bele-bele | Covid | Ha-grid | LEMB-Pass-Key | Legal-Bench | MIR-ACL | MLQA | Sci-docs | Spar-tQA | Stack-Over-flow | Stat-can-dial | TREC-COVID | Temp-Reas-onL1 | Twi-tter | Wiki | WinoG |
|---|---|---|---|---|---|---|---|---|---|---|---|---|---|---|---|---|---|---|---|
| BGE$_{multilingual}$[a] | 53.19 | 38.47 | 77.37 | 77.46 | 63.02 | 91.96 | 36.00 | 84.48 | 52.57 | 36.88 | 26.93 | 1.63 | 92.93 | 35.56 | 64.27 | 0.68 | 76.90 | 66.19 | 34.16 |
| NV-Embed-v2 | 56.71 | 31.58 | 70.07 | 69.79 | 59.21 | 98.90 | 42.75 | 95.20 | 55.54 | 70.61 | 21.90 | 11.51 | 92.93 | 19.55 | 88.85 | 6.50 | 45.57 | 90.61 | 49.74 |
| GritLM-7B | 58.27 | 41.80 | 63.17 | 71.70 | 73.40 | 98.67 | 38.25 | 94.99 | 51.68 | 72.74 | 24.41 | 9.35 | 93.37 | 45.76 | 74.31 | 7.16 | 43.27 | 91.05 | 53.70 |
| E5$_{mistral-7b}$ | 55.71 | 34.54 | 61.65 | 68.29 | 73.11 | 98.80 | 30.75 | 94.60 | 58.03 | 67.07 | 16.31 | 10.04 | 91.02 | 44.83 | 87.03 | 3.60 | 33.41 | 90.26 | 39.51 |
| GTE$_{Qwen2-7B}$ | 60.01 | 33.77 | 54.57 | 77.54 | 81.04 | 99.05 | 38.50 | 95.21 | 51.58 | 78.69 | 23.48 | 18.78 | 84.35 | 37.87 | 80.37 | 2.18 | 68.64 | 87.73 | 66.81 |
| **Lychee-embed$_{Qwen2.5-7B}$** | 66.61 | 48.32 | 87.50 | 77.57 | 81.47 | 98.82 | 39.10 | 95.82 | 63.75 | 70.03 | 38.84 | 29.46 | 96.40 | 48.32 | 63.80 | 5.20 | 89.91 | 90.01 | 74.59 |
| **Lychee-rerank$_{Qwen2.5-7B}$** | 68.77 | 53.10 | 80.38 | 78.30 | 92.32 | 98.90 | 40.01 | 95.53 | 66.25 | 72.77 | 38.29 | 37.16 | 95.54 | 49.10 | 73.45 | 9.28 | 89.92 | 92.10 | 75.71 |
| GTE$_{Qwen2-1.5B}$ | 60.76 | 34.89 | 69.72 | 66.59 | 80.79 | 98.69 | 38.25 | 94.84 | 63.23 | 72.89 | 24.98 | 18.63 | 90.27 | 33.25 | 85.38 | 1.33 | 67.01 | 87.31 | 65.61 |
| **Lychee-embed$_{Qwen2.5-1.5B}$** | 59.28 | 35.03 | 69.28 | 63.21 | 78.56 | 98.79 | 38.60 | 93.55 | 58.10 | 60.45 | 26.41 | 8.90 | 93.78 | 32.27 | 59.17 | 0.61 | 62.56 | 87.35 | 38.97 |
| **Lychee-rerank$_{Qwen2.5-1.5B}$** | 62.47 | 38.64 | 69.15 | 64.41 | 90.83 | 98.16 | 38.50 | 94.33 | 65.51 | 68.30 | 29.97 | 36.92 | 95.00 | 39.14 | 70.03 | 4.67 | 69.86 | 90.23 | 60.43 |
| BGE-M3 (Dense) | 59.54 | 52.58 | 79.11 | 60.35 | 41.79 | -3.11 | 20.10 | 80.76 | 62.79 | 54.60 | 74.12 | | | | | | | | |
| Jina-v3 | 55.13 | 32.77 | 43.29 | 73.41 | 78.55 | 98.69 | 38.00 | 93.43 | 62.58 | 73.39 | 19.92 | 0.65 | 90.79 | 39.20 | 77.74 | 0.58 | 73.02 | 77.81 | 18.57 |
| **Lychee-embed$_{Qwen2.5-0.5B}$** | 55.71 | 36.57 | 79.00 | 55.72 | 75.67 | 98.87 | 38.00 | 93.67 | 52.09 | 49.54 | 22.99 | 11.84 | 90.96 | 22.92 | 64.04 | 1.06 | 50.43 | 81.35 | 52.53 |
| **Lychee-rerank$_{Qwen2.5-0.5B}$** | 58.09 | 38.05 | 60.30 | 65.76 | 90.92 | 98.16 | 38.50 | 94.12 | 59.57 | 62.31 | 21.52 | 21.52 | 90.01 | 36.23 | 79.28 | 2.73 | 46.67 | 89.64 | 49.73 |

Table 14: The retrieval performance on MMTEB (Enevoldsen et al., 2025). We report nDCG@10 scores.

| | Max Length | Avg. | ar | de | en | es | fr | hi | it | ja | ko | pt | ru | th | zh |
|---|---|---|---|---|---|---|---|---|---|---|---|---|---|---|---|
| BGE$_{multilingual}$ | 8192 | 49.10 | 41.97 | 42.59 | 44.89 | 75.97 | 69.27 | 34.80 | 58.97 | 50.83 | 35.88 | 74.14 | 55.01 | 32.14 | 21.89 |
| GTE$_{Qwen2-7B}$ | 32768 | 56.53 | 53.10 | 53.91 | 51.45 | 78.96 | 78.23 | 35.81 | 67.15 | 53.55 | 47.76 | 78.66 | 63.88 | 39.22 | 31.70 |
| **Lychee-embed$_{Qwen2.5-7B}$** | 32768 | 57.90 | 57.14 | 53.95 | 57.69 | 84.00 | 74.28 | 31.83 | 69.67 | 55.18 | 48.48 | 81.05 | 67.49 | 38.89 | 33.30 |
| **Lychee-rerank$_{Qwen2.5-7B}$** | 32768 | 64.50 | 67.23 | 59.15 | 72.46 | 88.19 | 83.36 | 34.48 | 74.10 | 61.42 | 58.43 | 84.43 | 71.37 | 38.71 | 45.65 |
| GTE$_{Qwen2-1.5B}$ | 32768 | 52.11 | 50.07 | 50.91 | 52.32 | 78.67 | 73.79 | 23.14 | 64.19 | 48.70 | 42.35 | 74.40 | 61.36 | 32.53 | 26.04 |
| **Lychee-embed$_{Qwen2.5-1.5B}$** | 32768 | 53.85 | 49.58 | 51.15 | 56.11 | 78.77 | 71.26 | 24.61 | 65.29 | 51.14 | 46.91 | 79.05 | 63.04 | 34.34 | 29.84 |
| **Lychee-rerank$_{Qwen2.5-1.5B}$** | 32768 | 64.09 | 64.04 | 57.19 | 61.80 | 87.01 | 83.27 | 33.50 | 75.62 | 63.88 | 59.01 | 85.31 | 75.65 | 40.44 | 46.54 |
| BGE-M3 (Dense) | 8192 | 52.50 | 47.6 | 46.1 | 48.9 | 74.8 | 73.8 | 40.7 | 62.7 | 50.9 | 42.9 | 74.4 | 59.5 | 33.6 | 26.0 |
| Jina-v3 | 8192 | 40.71 | 34.39 | 38.19 | 29.10 | 62.05 | 59.85 | 25.37 | 53.82 | 38.45 | 32.34 | 63.29 | 49.51 | 25.55 | 17.20 |
| **Lychee-embed$_{Qwen2.5-0.5B}$** | 32768 | 51.44 | 45.67 | 47.08 | 51.91 | 76.24 | 72.41 | 22.08 | 63.28 | 49.98 | 42.11 | 75.56 | 61.18 | 35.22 | 26.10 |
| **Lychee-rerank$_{Qwen2.5-0.5B}$** | 32768 | 59.74 | 59.86 | 53.85 | 68.43 | 81.82 | 82.04 | 27.82 | 69.09 | 57.02 | 52.51 | 83.17 | 64.20 | 35.02 | 41.87 |

Table 15: Evaluation of multilingual long-doc retrieval on the MLDR (Chen et al., 2024) testset (measured by nDCG@10).

| MTEB(Code) | Avg. | Apps | COIR-CodeSearch-Net | Code-Edit-Search | Code-Feedback-MT | Code-Feedback-ST | Code-SearchNet-CCR | Code-SearchNet | Code-Trans-Ocean-Contest | Code-Trans-Ocean-DL | CosQA | Stack-Overflow-QA | Synthetic-Text2SQL |
|---|---|---|---|---|---|---|---|---|---|---|---|---|---|
| BGE$_{multilingual}$ | 62.04 | 22.93 | 68.14 | 60.48 | 60.52 | 76.70 | 73.23 | 83.43 | 86.84 | 32.64 | 27.93 | 92.93 | 58.67 |
| NV-Embed-v2 | 63.74 | 29.72 | 61.85 | 73.96 | 60.27 | 81.72 | 68.82 | 86.61 | 89.14 | 33.40 | 34.82 | 92.36 | 60.90 |
| GTE$_{Qwen2-7B}$ | 62.17 | 28.39 | 71.79 | 67.06 | 57.66 | 85.15 | 66.24 | 86.96 | 81.83 | 32.17 | 31.26 | 84.30 | 53.22 |
| **Lychee-embed$_{Qwen2.5-7B}$** | 79.93 | 93.20 | 89.70 | 79.50 | 94.58 | 89.48 | 88.83 | 92.54 | 94.37 | 36.97 | 40.59 | 96.40 | 63.05 |
| **Lychee-rerank$_{Qwen2.5-7B}$** | 80.78 | 94.85 | 90.93 | 82.87 | 95.90 | 90.09 | 97.00 | 91.96 | 94.67 | 36.97 | 37.63 | 95.54 | 62.93 |
| GTE$_{Qwen2-1.5B}$ | 61.98 | 28.91 | 71.56 | 59.60 | 49.92 | 81.92 | 72.08 | 91.00 | 79.02 | 32.73 | 32.23 | 90.27 | 54.49 |
| **Ours$_{Qwen2.5-1.5B}$** | 72.54 | 85.74 | 80.11 | 76.18 | 90.71 | 95.86 | 86.70 | 89.86 | 85.70 | 34.04 | 31.22 | 93.78 | 60.70 |
| **Lychee-rerank$_{Qwen2.5-1.5B}$** | 78.03 | 74.14 | 88.96 | 77.63 | 83.71 | 83.91 | 94.01 | 91.43 | 84.73 | 33.13 | 37.64 | 95.01 | 62.89 |
| BGE-M3 (Dense) | 58.22 | 14.77 | 58.07 | 59.83 | 47.86 | 69.27 | 53.55 | 86.22 | 26.29 | 27.36 | | 80.71 | 49.65 |
| Jina-v3 | 58.85 | 28.99 | 67.83 | 57.24 | 59.66 | 78.13 | 54.17 | 85.50 | 77.37 | 30.91 | 35.15 | 90.79 | 41.49 |
| **Lychee-embed$_{Qwen2.5-0.5B}$** | 71.06 | 68.16 | 35.64 | 60.0 | 88.70 | 85.70 | 80.21 | 88.86 | 82.02 | 33.98 | 38.33 | 88.96 | 60.63 |
| **Lychee-rerank$_{Qwen2.5-0.5B}$** | 69.86 | 61.52 | 85.11 | 72.53 | 73.01 | 84.67 | 90.49 | 89.25 | 70.84 | 25.37 | 34.75 | 90.16 | 60.76 |

Table 16: Performance on MTEB(Code) (Enevoldsen et al., 2025). We report nDCG@10 scores.

| FollowIR | Avg. | Core17 | News21 | Robust04 |
|---|---|---|---|---|
| BGE$_{multilingual}$[γ] | -2.13 | -2.49 | -0.29 | - 3.49 |
| NV-Embed-v2 | 1.04 | 3.21 | 1.58 | -1.67 |
| GritLM-7B | 3.45 | 6.70 | 1.22 | 2.44 |
| E5$_{mistral-7b}$ | -0.62 | 3.76 | 0.74 | -6.35 |
| GTE$_{Qwen2-7B}$ | 4.94 | 6.78 | 4.11 | 3.93 |
| Lychee-embed$_{Qwen2.5-7B}$ | 6.91 | 8.50 | 5.71 | 6.55 |
| **Lychee-rerank$_{Qwen2.5-7B}$** | 12.11 | 15.70 | 16.29 | 4.35 |
| GTE$_{Qwen2-1.5B}$ | 0.74 | 2.84 | 2.95 | -3.55 |
| Lychee-embed$_{Qwen2.5-1.5B}$ | 5.74 | 8.60 | 4.88 | 3.73 |
| **Lychee-rerank$_{Qwen2.5-1.5B}$** | 7.38 | 9.68 | 6.97 | 4.73 |
| BGE-M3 (Dense) | -3.11 | -1.25 | -1.39 | -6.69 |
| Jina-v3 | -1.34 | -0.06 | 2.36 | -6.31 |
| **Ours$_{Qwen2.5-0.5B}$** | 2.24 | 3.38 | 2.12 | 1.22 |
| **Lychee-rerank$_{Qwen2.5-0.5B}$** | 6.29 | 9.58 | 6.45 | 2.85 |

Table 17: The retrieval performance on FollowIR (Weller et al., 2025a). We report p-MRR scores.

| | StackExchange | | | | | | | Coding | | Theorem-based | | | Avg. |
|---|---|---|---|---|---|---|---|---|---|---|---|---|---|
| | Bio. | Earth. | Econ. | Psy. | Rob. | Stack. | Sus. | Leet. | Pony | aops | TheoQ. | TheoT. | |
| BGE$_{\text{multilingual}}$$^{\gamma}$ | 16.42 | 23.13 | 15.44 | 23.62 | 14.19 | 14.2 | 25.96 | 17.43 | 14.38 | 9.52 | 23.43 | 14.49 | 17.68 |
| NV-Embed-v2$^{\gamma}$ | 22.59 | 33.38 | 23.45 | 26.72 | 18.29 | 21.20 | 19.17 | 27.52 | 6.0 | 9.32 | 16.77 | 6.91 | 19.28 |
| GritLM-7B | 25.04 | 32.77 | 19.0 | 19.93 | 17.31 | 11.62 | 18.04 | 29.85 | 21.98 | 8.91 | 23.34 | 19.75 | 20.63 |
| E5$_{\text{mistral-7b}}$ | 18.84 | 25.96 | 15.49 | 15.79 | 16.37 | 9.83 | 18.52 | 28.72 | 4.81 | 7.1 | 23.94 | 25.09 | 17.54 |
| GTE$_{\text{Qwen2-7B}}$ | 32.09 | 40.66 | 16.18 | 26.58 | 12.82 | 13.95 | 20.82 | 31.07 | 1.25 | 15.1 | 29.9 | 34.22 | 22.89 |
| **Lychee-embed$_{\text{Qwen2.5-7B}}$** | 28.05 | 34.94 | 23.27 | 28.58 | 14.41 | 14.46 | 29.01 | 7.9 | 18.44 | 13.52 | 39.69 | 41.53 | 24. |
| **Lychee-rerank$_{\text{Qwen2.5-7B}}$** | 28.79 | 34.49 | 23.16 | 28.08 | 18.45 | 14.99 | 29.22 | 8.22 | 18.93 | 18.08 | 39.74 | 41.68 | 25.31 |
| GTE$_{\text{Qwen2-1.5B}}$$^{\gamma}$ | 17.56 | 29.38 | 25.14 | 24.8 | 10.38 | 10.8 | 26.44 | 7.07 | 16.39 | 14.87 | 20.1 | 18.73 | 18.47 |
| **Lychee-embed$_{\text{Qwen2.5-1.5B}}$** | 18.45 | 30.46 | 26.25 | 26.7 | 11.67 | 11.99 | 27.07 | 7.12 | 17.54 | 11.55 | 22.87 | 21.97 | 19.47 |
| **Lychee-rerank$_{\text{Qwen2.5-1.5B}}$** | 17.76 | 19.12 | 10.64 | 13.89 | 15.47 | 11.62 | 24.65 | 2.13 | 13.15 | 16.71 | 25.69 | 32.53 | 16.92 |
| BGE-M3$^{\gamma}$ (Dense) | 9.47 | 15.26 | 11.71 | 13.42 | 13.05 | 9.23 | 10.12 | 24.79 | 14.83 | 4.56 | 12.62 | 4.25 | 11.94 |
| Jina-v3$^{\gamma}$ | 13.47 | 11.46 | 14.48 | 8.67 | 11.57 | 5.93 | 8.83 | 18.07 | 3.97 | 13.47 | 22.79 | 16.91 | 11.34 |
| **Lychee-embed$_{\text{Qwen2.5-0.5B}}$** | 8.35 | 13.20 | 10.21 | 8.17 | 11.46 | 8.53 | 21.16 | 12.57 | 10.26 | 13.87 | 22.57 | 18.46 | 13.25 |
| **Lychee-rerank$_{\text{Qwen2.5-0.5B}}$** | 8.52 | 12.24 | 6.61 | 8.18 | 9.97 | 5.03 | 18.65 | 4.26 | 6.91 | 13.17 | 21.58 | 23.86 | 11.58 |

Table 18: The retrieval performance on BRIGHT (Su et al., 2025). We report nDCG@10 scores.

| Task Name | Instruction Template |
|---|---|
| BornholmBitextMining | Retrieve parallel sentences. |
| BibleNLPBitextMining | Retrieve parallel sentences |
| BUCC.v2 | Retrieve parallel sentences |
| DiaBlaBitextMining | Retrieve parallel sentences |
| FloresBitextMining | Retrieve parallel sentences |
| IN22GenBitextMining | Retrieve parallel sentences |
| IndicGenBenchFloresBitextMining | Retrieve parallel sentences |
| NollySentiBitextMining | Retrieve parallel sentences |
| NorwegianCourtsBitextMining | Retrieve parallel sentences in Norwegian Bokmål and Nynorsk |
| NTREXBitextMining | Retrieve parallel sentences |
| NusaTranslationBitextMining | Retrieve parallel sentences |
| NusaXBitextMining | Retrieve parallel sentences |
| Tatoeba | Retrieve parallel sentences |
| BulgarianStoreReviewSentimentClassfication | Classify user reviews into positive or negative sentiment |
| CzechProductReviewSentimentClassification | Classify product reviews into positive or negative sentiment |
| GreekLegalCodeClassification | Given a greek legal text, classify its topic |
| DBpediaClassification | Given a Wikipedia articles, categorized it into classes based on its DBpedia ontology. |
| FinancialPhrasebankClassification | Given financial news, categorized by sentiment into positive, negative, or neutral |
| PoemSentimentClassification | Gvien a poem, categorized by sentiment into positive, no_impact, negative or mixed |
| ToxicConversationsClassification | Classify the given comments as either toxic or not toxic |
| TweetTopicSingleClassification | Gvien a twitter, classify its topic |
| EstonianValenceClassification | Given a news, classify its topic |
| FilipinoShopeeReviewsClassification | Gvien a shopreview, classify its type |
| GujaratiNewsClassification | Given a Gujarati news articles, classify ist topic |
| SentimentAnalysisHindi | Given a hindi text, categorized by sentiment into positive, negative or neutral |
| IndonesianIdClickbaitClassification | Given an Indonesian news headlines, classify its into clickbait or non-clickbait |
| ItaCaseholdClassification | Given a judgments, classify its topic |
| KorSarcasmClassification | Given a twitter, categorized it into sarcasm or not_sarcasm |

*Continued on next page*

| Task Name | Instruction Template |
| --- | --- |
| KurdishSentimentClassification | Given a text, categorized by sentiment into positive or negative |
| MacedonianTweetSentimentClassification | Given a Macedonian tweet, categorized by sentiment into positive, negative, or neutral |
| AfriSentiClassification | Given a text, categorized by sentiment into positive, negative, or neutral |
| AmazonCounterfactualClassification | Classify a given Amazon customer review text as either counterfactual or not-counterfactual |
| CataloniaTweetClassification | Given a tweet, categorized by sentiment into AGAINST, FAVOR or NEUTRAL |
| CyrillicTurkicLangClassification | Given a text, classify its language |
| IndicLangClassification | Given a text, classify its language |
| MasakhaNEWSClassification | Classify the News in the given texts into one of the seven category: politics, sports, health, business, entertainment, technology, religion |
| MassiveIntentClassification | Given a user utterance as query, find the user intents |
| MultiHateClassification | Given a text, categorized by sentiment into hate or non-hate |
| NordicLangClassification | Classify texts based on language |
| NusaParagraphEmotionClassification | Given a paragraph, classify its emotion |
| NusaX-senti | Given a text, categorized by sentiment into positive or negative |
| ScalaClassification | Classify passages in Scandinavian Languages based on linguistic acceptability |
| SwissJudgementClassification | Given a text, categorized it into approval or dismissal |
| NepaliNewsClassification | Given a new, categorized it into business, entertainment or sports |
| OdiaNewsClassification | Given a new, categorized it into business, entertainment or sports |
| PunjabiNewsClassification | Given a new, categorized it into two-classes |
| PolEmo2.0-OUT | Classify the sentiment of out-of-domain (products and school) online reviews |
| PAC | Classify the sentence into one of the two types: "BEZPIECZNE_POSTANOWIENIE_UMOWNE" and "KLAUZULA_ABUZYWNA" |
| SinhalaNewsClassification | Given a new, categorized it into political, business, technology, sports and Entertainment |
| CSFDSKMovieReviewSentimentClassification | Given a movie review, categorized it by its sentiment |
| SiswatiNewsClassification | Given a new, classify its topic |
| SlovakMovieReviewSentimentClassification | Given a movie review, categorized it into positive or negative |
| SwahiliNewsClassification | Given a new, classify its domain |
| DalajClassification | Classify texts based on linguistic acceptability in Swedish |
| TswanaNewsClassification | Given a new, classify its topic |
| IsiZuluNewsClassification | Given a new, classify its topic |
| WikiCitiesClustering | Identify of Wikipedia articles of cities by country |
| MasakhaNEWSClusteringS2S | Identify the topic or theme of the given news articles based on the titles |
| RomaniBibleClustering | Identify verses from the Bible in Kalderash Romani by book. |
| ArXivHierarchicalClusteringP2P | Identify the main and secondary category of Arxiv papers based on the titles and abstracts |
| ArXivHierarchicalClusteringS2S | Identify the main and secondary category of Arxiv papers based on the titles |
| BigPatentClustering.v2 | Identify the category of documents from the Big Patent dataset |

*Continued on next page*

| Task Name | Instruction Template |
| --- | --- |
| BiorxivClusteringP2P.v2 | Identify the main category of Biorxiv papers based on the titles and abstracts |
| MedrxivClusteringP2P.v2 | Identify the main category of Medrxiv papers based on the titles and abstracts |
| StackExchangeClustering.v2 | Identify the topic or theme of StackExchange posts based on the titles |
| HALClusteringS2S.v2 | Identify the topic of titles from HAL |
| SIB200ClusteringS2S | Identify the category of documents |
| WikiClusteringP2P.v2 | Identify the category of wiki passages |
| SNLHierarchicalClusteringP2P | Identify categories in a Norwegian lexicon |
| PlscClusteringP2P.v2 | Identify the category of titles+abstracts from Library of Science |
| SwednClusteringP2P | Identify news categories in Swedish passages |
| CLSClusteringP2P.v2 | Identify the main category of scholar papers based on the titles and abstracts |
| StackOverflowQA | Given a question about coding, retrieval code or passage that can solve user's question |
| TwitterHjerneRetrieval | Retrieve answers to questions asked in Danish tweets |
| ArguAna | Given a claim, find documents that refute the claim |
| SCIDOCS | Given a scientific paper title, retrieve paper abstracts that are cited by the given paper |
| SpartQA | Given the following spatial reasoning question, retrieve the right answer. |
| TempReasonL1 | Given the following question about time, retrieve the correct answer. |
| TRECCOVID | Given a query on COVID-19, retrieve documents that answer the query |
| WinoGrande | Given the following sentence, retrieve an appropriate answer to fill in the missing underscored part. |
| CovidRetrieval | Given a question on COVID-19, retrieve news articles that answer the question |
| KorHateSpeechMLClassification | Given a Korean online news comments, classify its fine-grained hate speech classes |
| MalteseNewsClassification | Given a maltese new, classify its topic |
| MultiEURLEXMultilabelClassification | Given a text, classify its topic |
| BrazilianToxicTweetsClassification | Given a tweet, classify its topic |
| CEDRClassification | Given a comment as query, find expressed emotions (joy, sadness, surprise, fear, and anger) |
| CTKFactsNLI | Retrieve semantically similar text |
| SprintDuplicateQuestions | Retrieve duplicate questions from Sprint forum |
| TwitterURLCorpus | Retrieve tweets that are semantically similar to the given tweet |
| ArmenianParaphrasePC | Retrieve semantically similar text |
| indonli | Retrieve semantically similar text |
| OpusparcusPC | Retrieve semantically similar text |
| PawsXPairClassification | Retrieve semantically similar text |
| RTE3 | Retrieve semantically similar text |
| XNLI | Retrieve semantically similar text |
| PpcPC | Retrieve semantically similar text |
| TERRa | Given a premise, retrieve a hypothesis that is entailed by the premise |
| AlloprofReranking | Given a question, retrieve passages that answer the question |
| VoyageMMarcoReranking | Given a Japanese search query, retrieve web passages that answer the question |
| RuBQReranking | Given a question, retrieve Wikipedia passages that answer the question. |
| T2Reranking | Given a Chinese search query, retrieve web passages that answer the question |
| GermanSTSBenchmark | Retrieve semantically similar text |

| Task Name | Instruction Template |
| --- | --- |
| SICK-R | Retrieve semantically similar text |
| STS12 | Retrieve semantically similar text |
| STS13 | Retrieve semantically similar text |
| STS14 | Retrieve semantically similar text |
| STSBenchmark | Retrieve semantically similar text |
| FaroeseSTS | Retrieve semantically similar text |
| FinParaSTS | Retrieve semantically similar text |
| JSICK | Retrieve semantically similar text |
| IndicCrosslingualSTS | Retrieve semantically similar text |
| SemRel24STS | Retrieve semantically similar text |
| STS17 | Retrieve semantically similar text |
| STS22.v2 | Retrieve semantically similar text |
| STSES | Retrieve semantically similar text |
| STSB | Retrieve semantically similar text |
| AmazonPolarityClassification | Classify Amazon reviews into positive or negative sentiment |
| AmazonReviewsClassification | Classify the given Amazon review into its appropriate rating category |
| ArguAna | Given a claim, find documents that refute the claim |
| ArxivClusteringP2P | Identify the main and secondary category of Arxiv papers based on the titles and abstracts |
| ArxivClusteringS2S | Identify the main and secondary category of Arxiv papers based on the titles |
| AskUbuntuDupQuestions | Retrieve duplicate questions from AskUbuntu forum |
| Banking77Classification | Given a online banking query, find the corresponding intents |
| BiorxivClusteringP2P | Identify the main category of Biorxiv papers based on the titles and abstracts |
| BiorxivClusteringS2S | Identify the main category of Biorxiv papers based on the titles |
| ClimateFEVER | Given a claim about climate change, retrieve documents that support or refute the claim |
| DBPedia | Given a query, retrieve relevant entity descriptions from DBPedia |
| EmotionClassification | Classify the emotion expressed in the given Twitter message into one of the six emotions: anger, fear, joy, love, sadness, and surprise |
| FEVER | Given a claim, retrieve documents that support or refute the claim |
| FiQA2018 | Given a financial question, retrieve user replies that best answer the question |
| HotpotQA | Given a multi-hop question, retrieve documents that can help answer the question |
| ImdbClassification | Classify the sentiment expressed in the given movie review text from the IMDB dataset |
| MTOPDomainClassification | Classify the intent domain of the given utterance in task-oriented conversation |
| MTOPIntentClassification | Classify the intent of the given utterance in task-oriented conversation |
| MassiveIntentClassification | Given a user utterance as query, find the user intents |
| MassiveScenarioClassification | Given a user utterance as query, find the user scenarios |
| MedrxivClusteringP2P | Identify the main category of Medrxiv papers based on the titles and abstracts |
| MedrxivClusteringS2S | Identify the main category of Medrxiv papers based on the titles |
| MindSmallReranking | Retrieve relevant news articles based on user browsing history |

*Continued on next page*

| Task Name | Instruction Template |
| --- | --- |
| NFCorpus | Given a question, retrieve relevant documents that best answer the question |
| NQ | Given a question, retrieve Wikipedia passages that answer the question |
| QuoraRetrieval | Given a question, retrieve questions that are semantically equivalent to the given question |
| RedditClustering | Identify the topic or theme of Reddit posts based on the titles |
| RedditClusteringP2P | Identify the topic or theme of Reddit posts based on the titles and posts |
| SCIDOCS | Given a scientific paper title, retrieve paper abstracts that are cited by the given paper |
| SICK-R | Retrieve semantically similar text |
| STS12 | Retrieve semantically similar text |
| STS13 | Retrieve semantically similar text |
| STS14 | Retrieve semantically similar text |
| STSBenchmark | Retrieve semantically similar text |
| SciDocsRR | Given a title of a scientific paper, retrieve the titles of other relevant papers |
| SciFact | Given a scientific claim, retrieve documents that support or refute the claim |
| SprintDuplicateQuestions | Retrieve duplicate questions from Sprint forum |
| StackExchangeClustering | Identify the topic or theme of StackExchange posts based on the titles |
| StackExchangeClusteringP2P | Identify the topic or theme of StackExchange posts based on the given paragraphs |
| StackOverflowDupQuestions | Retrieve duplicate questions from StackOverflow forum |
| TRECCOVID | Given a query on COVID-19, retrieve documents that answer the query |
| Touche2020 | Given a question, retrieve detailed and persuasive arguments that answer the question |
| ToxicConversationsClassification | Classify the given comments as either toxic or not toxic |
| TweetSentimentExtractionClassification | Classify the sentiment of a given tweet as either positive, negative, or neutral |
| TwentyNewsgroupsClustering | Identify the topic or theme of the given news articles |
| TwitterSemEval2015 | Retrieve tweets that are semantically similar to the given tweet |
| TwitterURLCorpus | Retrieve tweets that are semantically similar to the given tweet |
| MSMARCO | Given a web search query, retrieve relevant passages that answer the query |
| AmazonCounterfactualClassification | Classify a given Amazon customer review text as either counterfactual or not-counterfactual |
| STS17 | Retrieve semantically similar text |
| STS22 | Retrieve semantically similar text |
| T2Retrieval | Given a Chinese search query, retrieve web passages that answer the question |
| MMarcoRetrieval | Given a web search query, retrieve relevant passages that answer the query |
| DuRetrieval | Given a Chinese search query, retrieve web passages that answer the question |
| CovidRetrieval | Given a question on COVID-19, retrieve news articles that answer the question |
| CmedqaRetrieval | Given a Chinese community medical question, retrieve replies that best answer the question |
| EcomRetrieval | Given a user query from an e-commerce website, retrieve description sentences of relevant products |

| Task Name | Instruction Template |
| --- | --- |
| MedicalRetrieval | Given a medical question, retrieve user replies that best answer the question |
| VideoRetrieval | Given a video search query, retrieve the titles of relevant videos |
| T2Reranking | Given a Chinese search query, retrieve web passages that answer the question |
| MMarcoReranking | Given a Chinese search query, retrieve web passages that answer the question |
| CMedQAv1-reranking | Given a Chinese community medical question, retrieve replies that best answer the question |
| CMedQAv2-reranking | Given a Chinese community medical question, retrieve replies that best answer the question |
| Ocnli | Retrieve semantically similar text. |
| Cmnli | Retrieve semantically similar text. |
| CLSClusteringS2S | Identify the main category of scholar papers based on the titles |
| CLSClusteringP2P | Identify the main category of scholar papers based on the titles and abstracts |
| ThuNewsClusteringS2S | Identify the topic or theme of the given news articles based on the titles |
| ThuNewsClusteringP2P | Identify the topic or theme of the given news articles based on the titles and contents |
| LCQMC | Retrieve semantically similar text |
| PAWSX | Retrieve semantically similar text |
| AFQMC | Retrieve semantically similar text |
| QBQTC | Retrieve semantically similar text |
| TNews | Classify the fine-grained category of the given news title |
| IFlyTek | Given an App description text, find the appropriate fine-grained category |
| Waimai | Classify the customer review from a food takeaway platform into positive or negative |
| OnlineShopping | Classify the customer review for online shopping into positive or negative |
| JDReview | Classify the customer review for iPhone on e-commerce platform into positive or negative |
| MultilingualSentiment | Classify sentiment of the customer review into positive, neutral, or negative |
| BQ | Retrieve semantically similar text |
| STSB | Retrieve semantically similar text |
| MultilingualSentiment | Classify sentiment of the customer review into positive, neutral, or negative |
| AppsRetrieval | Given a question about coding, retrieval code or passage that can solve user's question |
| CodeEditSearchRetrieval | Given a question about coding, retrieval code or passage that can solve user's question |
| CodeFeedbackMT | Given a question about coding, retrieval code or passage that can solve user's question |
| CodeFeedbackST | Given a question about coding, retrieval code or passage that can solve user's question |
| CodeSearchNetCCRetrieval | Given a question about coding, retrieval code or passage that can solve user's question |
| CodeSearchNetRetrieval | Given a question about coding, retrieval code or passage that can solve user's question |
| CodeTransOceanContest | Given a question about coding, retrieval code or passage that can solve user's question |
| CodeTransOceanDL | Given a question about coding, retrieval code or passage that can solve user's question |
| CosQA | Given a question about coding, retrieval code or passage that can solve user's question |

| Task Name | Instruction Template |
|---|---|
| COIRCodeSearchNetRetrieval | Given a question about coding, retrieval code or passage that can solve user's question |
| StackOverflowQA | Given a question about coding, retrieval code or passage that can solve user's question |
| SyntheticText2SQL | Given a question about coding, retrieval code or passage that can solve user's question |
| AlloProfClusteringS2S.v2 | Identify the topic of document titles from Allo Prof dataset |
| AILAStatutes | Identifying the most relevant statutes for a given situation |
| HagridRetrieval | Retrieval the relevant passage for the given query |
| LegalBenchCorporateLobbying | Retrieval the relevant passage for the given query |
| LEMBPasskeyRetrieval | Retrieval the relevant passage for the given query |
| BelebeleRetrieval | Retrieval the relevant passage for the given query |
| MLQARetrieval | Retrieval the relevant passage for the given query |
| StatcanDialogueDatasetRetrieval | Retrieval the relevant passage for the given query |
| WikipediaRetrievalMultilingual | Retrieval the relevant passage for the given query |
| Core17InstructionRetrieval | Retrieval the relevant passage for the given query |
| News21InstructionRetrieval | Retrieval the relevant passage for the given query |
| Robust04InstructionRetrieval | Retrieval the relevant passage for the given query |
| WebLINXCandidatesReranking | Retrieval the relevant passage for the given query |
| WikipediaRerankingMultilingual | Retrieval the relevant passage for the given query |
| STS15 | Retrieve semantically similar text |
| MIRACLRetrievalHardNegatives | Retrieve semantically similar text |
| BIOSSES | Retrieve semantically similar text |
| CQADupstackRetrieval | Given a question, retrieve detailed question descriptions from Stackexchange that are duplicates to the given question |
| STS15 | Retrieve semantically similar text |
| STS16 | Retrieve semantically similar text |
| SummEval | Retrieve semantically similar text |
| ATEC | Retrieve semantically similar text |
| aops | Given a Math problem, retrieve relevant examples that help answer the problem |
| biology | Given a post, retrieve relevant passages that help answer the post |
| earth_science | Given a post, retrieve relevant passages that help answer the post |
| economics | Given a economics post, retrieve relevant passages that help answer the post |
| leetcode | Given a coding problem, retrieve relevant examples that help answer the problem |
| pony | Given a question about pony program language, retrieve relevant passages that helpanswer the question |
| psychology | Given a psychology post, retrieve relevant passages that help answer the post |
| theoremqa_questions | Given a Math problem, retrieve relevant examples that help answerthe problem |
| theoremqa_theorems | Given a Math problem, retrieve relevant theorems that help answer theproblem |
| robotics | Given a robotics post, retrieve relevant passages that help answer the post |
| stackoverflow | Given a stackoverflow post, retrieve relevant passages that help answerthe post |
| sustainable_living | Given a sustainable_living post, retrieve relevant passages that help answer the post |

Table 19: The instruction we used on the evaluation benchmarks.

