# OpenReview forum: "Phased Training for LLM-powered Text Retrieval Models Beyond Data Scaling"
_colmweb.org/COLM/2025/Conference — COLM 2025_

### Official Review · Reviewer_G5Ni · 2025-05-09

**Rating:** 7
**Confidence:** 5
**Ethics Flag:** 1

**Summary:**

This work proposes a phased training approach to be able to train a model that is successful at first stage retrieval and reranking for many domains. Rather than training all data for all tasks, the paper investigates first warming up the model on a sample of all tasks, then undertaking task learning, and finally merging the models to obtain a single model for retrieval. The tasks are basic relevance, tool retrieval, code retrieval, and instruction-based retrieval.

In addition to the phased training, the authors utilize synthetic data for training. Synthetic data is generated with a two-step approach. After identifying a document, a representative user is chosen from PersonaHub to describe the searcher. Attributes of the desired query are also specified include the type of query, the difficulty level, the length, and the language to generate the query in. Once the query is generated, it is discarded if it is formatted incorrectly or does not retrieve the source document among the top 20 results.

The approach is evaluated on 5 tasks including both monolingual retrieval and multilingual retrieval. Given the interest in multilingual retrieval, it is surprising that mFollowIR is not included.

The paper is well written and generally easy to follow. The approach demonstrates robustness over the various datasets, especially the largest Qwen model that was trained, Qwen2.5-7B.

A surprising result is how ineffective other models are at re-ranking the first-state retrieval result of the author's first stage retrieval (Table 3). Even the largest re-ranker used (BGE_reranker-v2-gemma) frequently performed worse than the run it was reranking. This makes the reader question whether weak baseline systems were selected or whether the documents ranked highly by this approach are substantially different from the approaches where those rerankers perform well.

When considering the ablations, the authors appear to brush over the naive approach with up-sampling, but it appears to perform as well as their approach (sometimes a bit better, sometimes a bit more depending on the dataset).   This really needs to be acknowledged in the text. In addition it is unclear if the up-sampling approach utilizes the synthetic data and where the cost of producing that data belongs when considering the training efficiency.

**Questions To Authors:**

Additional multilingual experiments over NeuCLIR (both news and Chinese technical documents) and well as mFollowIR would strengthen the multilingual results in the paper.

**Reasons To Accept:**

+ Phased training is a novel approach to training a robust first stage retriever
+ The use of personas is synthetic data generation is novel
+ A large variety of benchmarks were included to the results

**Reasons To Reject:**

- The reranking baselines appear to be very weak and likely inflate the gain in effectiveness of this approach
- The strong performance of up-sampling is overlooked in the prose

---

> ### Author Response · Authors · 2025-05-31
> **Response to G5Ni**
>
> ### W1
> > The reranking baselines appear to be very weak and likely inflate the gain in effectiveness of this approach
>
> We acknowledge this limitation. The choice of reranking baselines is constrained by the current availability of strong open-source reranking models. We will make effort to extend our comparisons with additional strong baselines.
>
> ---
>
> ### W2
> > The strong performance of up-sampling is overlooked in the prose
>
> Yes, simple up-sampling indeed achieves good performance and is widely adopted. Our approach demonstrates superior results with better training efficiency compared to up-sampling. We believe this advantage is significant and warrants further research from the community.
>
> ---
>
> ### Q1
> > Additional multilingual experiments over NeuCLIR (both news and Chinese technical documents) and well as mFollowIR would strengthen the multilingual results in the paper.
>
> Thank you for the valuable suggestion. We will incorporate these additional benchmarks.
>
> ---
>
> We greatly appreciate your thoughtful comments and valuable feedback. Thank you for helping us improve our work.

---

> > ### Comment · Reviewer_G5Ni · 2025-06-08
> >
> > I have read the response. I hope the camera ready paper will include the additional results. My score will remain the same.

---

> > > ### Author Response · Authors · 2025-06-10
> > >
> > > Thank you! Will do.

---

### Official Review · Reviewer_n97a · 2025-05-12

**Rating:** 7
**Confidence:** 4
**Ethics Flag:** 1

**Summary:**

The paper studies the problem of text retrieval across various domains. It aims to resolve challenges in data imbalance and efficiency in training retrieval models. The paper studying namely 5 retrieval tasks spanning relevance retrieval, code retrieval, tool retrieval, complex instruction, and reasoning-intensive retrieval. The paper proposes training the model in a phased manner, which involves:
1. Warming up: using equal-sized subsets of data.
2. Task learning, which is analogous to task-specific learning. This is done on the same model across tasks, creating multiple copies of the model.
3. Merge: the various merged models are then merged using the SLEPT merging technique into a single model.
4. Enhancement: This involves again training the model over the task-specific data from the warmup stage for inference.

The authors further also propose generating of synthetic data that can aid in improving the training data. They do this using various query types (keywords, facts, summarization, classification / yes/no , and background. They further set different discovery levels for the query and then make use of an LLM to synthesize the synthetic query data. They also filter the data and consider checks for format validation and consistency using an embedding model.

The authors provide emperical evidence of outperforming various other models that are in the same parameter count for both tasks, retrieval and reranking.

**Questions To Authors:**

The paper could improve on the writing, mainly explaining:

1. More on how the cost per training iteration is computed  in section 4.3
2. Including more details on the synthesis pipeline and statistics regaring question quality and across the various tasks.
3. Whar metric is used in table 2 ? Is it ndcg@10 ?
4. The main text should mention that Qwen-2.5-72B instruct was used for synthetic data generation.
5. There are various models that are trained on the MSMARCO set and evaluated on BIER, why do the authors just compare against the Jina, BGE (gemma and m3) on the reranking task?
6. What are the insights in the BGE with gemma outperforming in table 3 for the MMTEB-R subset.
7. For completeness, can the numbers in table 2 be filled in for the CMTEB MLDR, MTEB Code for NV-Embed-v2, GritLM-7B, and E5 mistral?
8. Given that the phased training setup is more efficient, for the same cost it would be interesting to see the naive baseline numbers.
9. It would be interesting to see what happens when sematic classes can be clustered together for phased training

Please also read the weaknesses for other concerns.

**Reasons To Accept:**

The problem studied is interesting and shows that phased training provides benefits as compared to other techniques. Further the authors also provide a synthetic dataset that is composed of 2M passages both in english and chinese which can be helpful towards the broader research community.

**Reasons To Reject:**

While the paper shows that the proposed technique outperforms the other studied baselines, it lacks the following:
1. comparison with a pretrained embeddings from the studied model (Qwen-2.5 0.5B, 1.5B, and 7B)
2. Further, it still remains to be seen if the technique generalizes over other families of models as well.
3. In Table 4, there is no discussion on how significant the improvements of the proposed phase training framework given that in many cases naive upsampling also does a good job.
5. The reranker often underperforms the embedding model (Table 3) and only the 7B reranker outperforms the embedder baseline.
6. Given that there is a factor of difficulty for queries, there could have been a small user study  / semi automated verification done towards accessing that the LLM produces varied difficulty of questions.
7. Quantitative statistics for the synthetic dataset (like number of tokens per query and other parameters) are missing, which may raise concerns about the synthetic data.

---

> ### Author Response · Authors · 2025-05-31
> **Response to n97a**
>
> ### W1
> > comparison with a pretrained embeddings from the studied model (Qwen-2.5 0.5B, 1.5B, and 7B)
>
> We have included comparisons with embeddings from the same model (1.5B) trained using standard methods in Table 4 (rows 6 and 7). We will extend the comparisons to include more Qwen2.5-based model variants in the revision.
>
> ---
>
> ### W2
> > Further, it still remains to be seen if the technique generalizes over other families of models as well.
>
> We agree with you and will make efforts to extend to other models, such as the LLAMA family.
>
> ---
>
> ### W3
> > In Table 4, there is no discussion on how significant the improvements of the proposed phase training framework given that in many cases naive upsampling also does a good job.
>
> You raise a insightful point about the writing. While Table 4 shows comparable performance between our method and upsampling, the key advantage lies in training efficiency, as discussed in Section 4.3. Our approach achieves slightly better performance in significantly less time (7 days vs. 13.5 days for upsampling).
>
> We will better highlight this discussion about efficiency-performance trade-off in the revision.
>
> ---
>
> ### W4
> > The reranker often underperforms the embedding model (Table 3) and only the 7B reranker outperforms the embedder baseline.
>
> Yes, this performance pattern is common when embedding models are strong while reranking models are not robust enough. Notably, our 1.5B reranker maintains performance parity with embeddings, demonstrating the effectiveness of our approach. The 7B variant further shows clear improvements.
>
> ---
>
> ### W5
> > Given that there is a factor of difficulty for queries, there could have been a small user study / semi automated verification done towards accessing that the LLM produces varied difficulty of questions.
>
> We thank the valuable suggestion and will conduct this experiment.
>
> ---
>
> ### W6
> > Quantitative statistics for the synthetic dataset (like number of tokens per query and other parameters) are missing, which may raise concerns about the synthetic data.
>
> We will add these statistics.
>
> ---
>
> ### Q1, Q2, Q4, Q7, Q9
> Thank you for the helpful comments, we will make changes accordingly.
>
> ---
>
> ### Q3
> > Whar metric is used in table 2 ? Is it ndcg@10 ?
>
> For MTEB, CMTEB, MMTEB, they consist of multiple tasks like retrieval, clustering, classification, each using different metrics. We follow their benchmark scoring methods and report the overall average.
> For MLDR, ToolBench, BRIGHT, we use nDCG@10.
> For FollowIR, it is a newly defined p-MRR metric.
>
> We will clarify these metric details in the revision.
>
> ---
>
> ### Q5
> > There are various models that are trained on the MSMARCO set and evaluated on BIER, why do the authors just compare against the Jina, BGE (gemma and m3) on the reranking task?
>
> Our evaluation goes beyond BEIR to include specialized retrieval scenarios like FollowIR and BRIGHT. We selected these models (Jina, BGE) as baselines because they are trained on broader datasets and are generally considered more versatile, making them strong baselines for our expanded evaluation scope.
>
> ---
>
> ### Q6
> > What are the insights in the BGE with gemma outperforming in table 3 for the MMTEB-R subset.
>
> We believe BGE with Gemma's better performance on MMTEB-R might be attributed to their superior multilingual training data, as it's specifically designed for multilingual retrieval. Additionally, the base model Gemma supports more languages than Qwen. We apologize for the bold number error and will fix it in revision.
>
> ---
>
> ### Q8
> > Given that the phased training setup is more efficient, for the same cost it would be interesting to see the naive baseline numbers.
>
> Thanks for the valuable suggestion. Indeed comparing performance under the same computational budget is also helpful to the understanding. We will add these results.
>
> ---
>
> We greatly appreciate your thoughtful comments and valuable feedback.
>
> If you have any additional questions or concerns, we welcome further discussion.
>
> If our responses have addressed your concerns, we respectfully invite you to consider increasing your rating score.
>
> Thank you for helping us improve our work.

---

> > ### Comment · Reviewer_n97a · 2025-06-03
> >
> > > We selected these models (Jina, BGE) as baselines because they are trained on broader datasets and are generally considered more versatile, making them strong baselines for our expanded evaluation scope.
> >
> > Could you please provide a leaderboard/citation towards this?

---

> > > ### Author Response · Authors · 2025-06-03
> > >
> > > For jina, the blog states it supports multilingual, code and other scenarios in reranking.
> > >
> > > https://jina.ai/news/jina-reranker-v2-for-agentic-rag-ultra-fast-multilingual-function-calling-and-code-search/
> > >
> > > For BGE modes, the model cards say that they are multilingual.
> > >
> > > https://huggingface.co/BAAI/bge-reranker-v2-m3
> > >
> > > https://huggingface.co/BAAI/bge-reranker-v2-gemma
> > >
> > > ---
> > >
> > > Anyway, we will expand more baselines.
> > >
> > > Thank you for having a discussion with us!

---

> > > > ### Comment · Reviewer_n97a · 2025-06-04
> > > >
> > > > Thank you for the discussion. I have increased my score following the discussion. Please make the promised changes as you have highlighted:
> > > > 1. Extending the comparisons to include more Qwen2.5-based model variants in the revision.
> > > > 2. Quantitative metric details in the revision.
> > > > 3. Comparing performance under the same computational budget
> > > > 4. A small user study / semi automated verification done towards accessing that the LLM produces varied difficulty of questions
> > > > 5. Highlight discussion about efficiency-performance trade-off in the revision
> > > > 6. Addressing Q1, Q2, Q4, Q7, Q9

---

> > > > > ### Author Response · Authors · 2025-06-05
> > > > >
> > > > > Thank you! Will work on these.

---

### Official Review · Reviewer_5b9A · 2025-05-13

**Rating:** 8
**Confidence:** 5
**Ethics Flag:** 1

**Summary:**

This paper introduces a novel "Phased Training Framework" and an efficient LLM-driven data synthesis pipeline to build general-purpose text retrieval models (embedding and reranking). The framework tackles multi-task conflict and data imbalance via a four-step process: warm-up, task learning, model merging, and enhancement. The methodology leads to state-of-the-art results on a wide variety of retrieval tasks for both embedding and reranking, contributing valuable techniques and practices to the field.

**Questions To Authors:**

- What is the reason for “we evaluate on the full set for embedding models and the retrieval subset for reranking”? How does reranking mode perform on non-retrieval tasks?
- Can you confirm if it is intended that in Eq4 that you sum up the weights of all task vectors instead of averaging?
- Could you elaborate on the specific methodology for clustering datasets based on "task similarity and data volume"?
- How were the SLERP merging hyperparameters (e.g., t, lambda) determined? Additionally, how does SLERP merging compare to simply averaging the model weights?

**Reasons To Accept:**

- Innovative Unified Architecture: A key contribution of this work is a novel model architecture that natively supports both embedding and reranking functionalities,
- Effective Phased Training: The proposed multi-stage training addresses multi-task learning challenges in retrieval.
- Efficient Data Synthesis: Offers a scalable, cost-effective method for high-quality synthetic data generation using open-source LLMs (Qwen-2.5-72B-Instruct).
- State-of-the-Art Results: Demonstrates superior performance across numerous benchmarks and task types.

**Reasons To Reject:**

- Incremental Novelty: The proposed framework largely combines existing techniques (multi-stage training, model merging, synthetic data), potentially lacking fundamental novelty beyond an engineering contribution.
- Insufficient Comparison Baselines: To showcase the advantage in terms of training efficiency, the paper compares mainly against simple "naïve" multi-task training and up-sampling, failing to benchmark against more sophisticated existing strategies, making superiority claims less convincing.

---

> ### Author Response · Authors · 2025-05-31
> **Response to 5b9A**
>
> ### W1
> > Incremental Novelty: The proposed framework largely combines existing techniques (multi-stage training, model merging, synthetic data), potentially lacking fundamental novelty beyond an engineering contribution.
>
> We agree with your observation. But our work's core contribution is exploring an efficient pipeline for constructing LLM-based retrieval models. We show that effectively integrating existing techniques is able to address the problem at hand.
>
> ---
>
> ### W2
> > Insufficient Comparison Baselines: To showcase the advantage in terms of training efficiency, the paper compares mainly against simple "naïve" multi-task training and up-sampling, failing to benchmark against more sophisticated existing strategies, making superiority claims less convincing.
>
> While up-sampling represents the most widely adopted approach in current embedding and reranking models, we acknowledge the need to compare with more sophisticated methods. We will expand our comparisons in the revised version.
>
> ---
>
> ### Q1
> > What is the reason for “we evaluate on the full set for embedding models and the retrieval subset for reranking”? How does reranking mode perform on non-retrieval tasks?
>
> Reranking models are *specifically designed for retrieval* tasks, making them less directly applicable to other NLP tasks such as classification or clustering. Therefore, we focused our evaluation of reranking models on the retrieval subset.
>
> In contrast, embedding models are known for their *task-agnostic* nature and versatility across different NLP applications. For instance, embeddings can be effectively combined with linear classifiers for text classification tasks, or used for semantic similarity measurements in various scenarios. Following standard practices in the field, we evaluated embedding models on comprehensive benchmarks like MTEB, which covers a wide range of tasks and better reflects their general-purpose capabilities.
>
> ---
>
> ### Q2
> > Can you confirm if it is intended that in Eq4 that you sum up the weights of all task vectors instead of averaging
>
> Yes, this is the weighed sum. See Formula 3 for the weighting method.
>
> ---
>
> ### Q3
> > Could you elaborate on the specific methodology for clustering datasets based on "task similarity and data volume"?
>
> The clustering is conducted through manual categorization, taking into account both task characteristics and data distribution patterns. Specifically:
> - Basic Retrieval (English & Chinese): These form the largest cluster, as they share similar fundamental retrieval patterns and represent the majority of our dataset volume.
> - Multilingual Retrieval: Grouped as a separate cluster due to its unique challenges and specific requirements in handling cross-lingual content.
> - Complex Instruction-based Retrieval: Isolated as a distinct cluster given its specialized nature involving instructions and more sophisticated query understanding.
> - Code & Tool Retrieval: Combined into a single cluster due to their similar data characteristics, both primarily focusing on code-centric content and sharing comparable structural patterns.
>
> This categorization allows us to better analyze model performance across different retrieval scenarios while maintaining meaningful task relationships and practical data handling considerations.
>
> ---
>
> ### Q4
> > How were the SLERP merging hyperparameters (e.g., t, lambda) determined? Additionally, how does SLERP merging compare to simply averaging the model weights?
>
> We employ a small scale optimization to search the hyperparameters, refer to section 3.3. In our preliminary experiments, SLERP demonstrates better balanced multi-task capabilities compared to simple averaging. We will add this comparative experiment in the next version paper.
>
> ---
>
> We greatly appreciate your thoughtful comments and valuable feedback. Thank you for helping us improve our work.

---

> > ### Comment · Reviewer_5b9A · 2025-06-08
> >
> > To follow up on my questions:
> > Q3: thanks for the clarification. Can you please highlight the "manual" nature of this method in the revised version? It is not clear to me initially.
> > Q4: Please provide more comparison results on different methods of model merging in the revised version to help readers understand its effects.

---

> > > ### Author Response · Authors · 2025-06-10
> > >
> > > Thank you! Will do.

---

### Official Review · Reviewer_uJyM · 2025-05-27

**Rating:** 7
**Confidence:** 4
**Ethics Flag:** 1

**Summary:**

This paper presents a four-stage training framework for adapting LLM-based text retrieval to diverse tasks, including relevance, code, and tool retrieval. It further introduces a role-guided data synthesis method to enhance retrieval performance. Experiments on eight benchmarks across five task types, using Qwen2.5 models (0.5B, 1.5B, 7B), demonstrate better performance over baseline methods in both retrieval and reranking. Ablation studies also show the effectiveness of each training stage and the data synthesis approach.

**Questions To Authors:**

1. Using the number of training iterations to estimate training efficiency is somewhat unconvincing, as it’s not very intuitive. I suggest using GPU time or resource usage as a more accurate and meaningful metric.

2. Do the naive baselines in Table 4 also use the same synthetic data? It would be helpful to include results for the baselines with the synthetic data incorporated for a fair comparison.

**Reasons To Accept:**

1. The paper is clearly written with well-designed and detailed experiments, reflecting the authors' strong efforts in advancing open-source work.

2. The proposed methods demonstrate strong performance improvements and are easy to implement. They are likely to be valuable to the community.

**Reasons To Reject:**

The proposed multi-stage training approach—comprising Warm-up, Task Learning, Merge, Enhance, and data synthesis—is not entirely novel, but the authors effectively integrate these components into a cohesive text retrieval pipeline.

---

> ### Author Response · Authors · 2025-05-31
> **Response to uJyM**
>
> ### W1
> > The proposed multi-stage training approach—comprising Warm-up, Task Learning, Merge, Enhance, and data synthesis—is not entirely novel, but the authors effectively integrate these components into a cohesive text retrieval pipeline.
>
> Yes, our primary contribution lies in proposing an effective and practical pipeline for efficiently building retrieval models, rather than the novelty of any single component.
>
> ---
>
> ### Q1
> > Using the number of training iterations to estimate training efficiency is somewhat unconvincing, as it’s not very intuitive. I suggest using GPU time or resource usage as a more accurate and meaningful metric.
>
> We appreciate this valuable comment. GPU time is indeed a more intuitive and accurate metric and we will include detailed statistics in the revised paper.
>
> As a quick response, here are preliminary GPU time info for the 1.5B embedding model trained on 32 GPUs:
>  - Our method: ~7 days
>  - Native method: ~11.5 days
>  - Up-sampling method: ~13.5 days
>
> This demonstrates our method's significant efficiency advantage in terms of actual training time.
>
> ---
>
> ### Q2
> > Do the naive baselines in Table 4 also use the same synthetic data? It would be helpful to include results for the baselines with the synthetic data incorporated for a fair comparison.
>
> Yes, two naive baselines also use the same data. We will provide more clear notations in the table.
>
> ---
>
> We greatly appreciate your thoughtful comments and valuable feedback. Thank you for helping us improve our work.

---

> > ### Comment · Reviewer_uJyM · 2025-06-03
> >
> > Looking forward to the open-source release.

---

### Decision · Program_Chairs · 2025-07-08

**Decision:**

Accept

**Comment:**

The authors propose a method for training a retriever that can handle a disparate array of retrieval tasks: basic relevance retrieval, code retrieval, tool retrieval, complex instruction based retrieval, and reasoning-intensive retrieval. To do so, they propose a "phased training" approach that entails training separate retrievers for each retrieval task and then merging the weights. The authors show that this outperforms baselines and they conduct an ablation study to tease out the source of the improvements. All reviewers appreciated the paper and recommended acceptance. I encourage the authors to take the reviewers' feedback into account for their next iteration, including all of the promised changes and in particular expanding the baselines and clarifying the writing. There have also been some recent retrievers that are not included in the paper (reasonably so, because of the concurrent submission policy) but could be compared to in the next version.